# Associations of ABO and Rhesus D blood groups with phenome-wide disease incidence: A 41-year retrospective cohort study of 482,914 patients

Peter Bruun-Rasmussen[1,2], Morten Hanefeld Dziegiel[1], Karina Banasik[2], Pär Ingemar Johansson[1], Søren Brunak[2]*

[1]Department of Clinical Immunology, Copenhagen University Hospital, Copenhagen, Denmark; [2]Novo Nordisk Foundation Center for Protein Research, University of Copenhagen, Copenhagen, Denmark

## Abstract

**Background:** Whether natural selection may have attributed to the observed blood group frequency differences between populations remains debatable. The ABO system has been associated with several diseases and recently also with susceptibility to COVID-19 infection. Associative studies of the RhD system and diseases are sparser. A large disease-wide risk analysis may further elucidate the relationship between the ABO/RhD blood groups and disease incidence.

**Methods:** We performed a systematic log-linear quasi-Poisson regression analysis of the ABO/RhD blood groups across 1,312 phecode diagnoses. Unlike prior studies, we determined the incidence rate ratio for each individual ABO blood group relative to all other ABO blood groups as opposed to using blood group O as the reference. Moreover, we used up to 41 years of nationwide Danish follow-up data, and a disease categorization scheme specifically developed for diagnosis-wide analysis. Further, we determined associations between the ABO/RhD blood groups and the age at the first diagnosis. Estimates were adjusted for multiple testing.

**Results:** The retrospective cohort included 482,914 Danish patients (60.4% females). The incidence rate ratios (IRRs) of 101 phecodes were found statistically significant between the ABO blood groups, while the IRRs of 28 phecodes were found statistically significant for the RhD blood group. The associations included cancers and musculoskeletal-, genitourinary-, endocrinal-, infectious-, cardiovascular-, and gastrointestinal diseases.

**Conclusions:** We found associations of disease-wide susceptibility differences between the blood groups of the ABO and RhD systems, including cancer of the tongue, monocytic leukemia, cervical cancer, osteoarthrosis, asthma, and HIV- and hepatitis B infection. We found marginal evidence of associations between the blood groups and the age at first diagnosis.

**Funding:** Novo Nordisk Foundation and the Innovation Fund Denmark

*For correspondence: soren.brunak@cpr.ku.dk

## Editor's evaluation

This important retrospective analysis of nearly 500,000 hospitalized Danish patients sheds light on the possible relationships between blood type and susceptibility to a host of diseases. The Danish National Patient Register is a compelling data source, and the statistical methodology is solid. The findings reported herein provide evidence, supporting information, and potential hypotheses for researchers interested in the causes and etiology of diseases as they relate to blood type.

## Introduction

Still 100 years after the discovery of the ABO and Rhesus D (RhD) blood group systems, the selective forces that may have attributed to the observed blood group population differences remain elusive (*Anstee, 2010*). The pathophysiological mechanisms behind the observed relationship between blood groups and diseases are not well understood either. The ABO system has been associated with susceptibility to multiple diseases, including gastrointestinal- and cardiovascular diseases and pancreatic-, gastric-, and ovarian cancers (*Vasan et al., 2016*; *Liumbruno and Franchini, 2014*; *Wolpin et al., 2010*; *Groot et al., 2020*; *Edgren et al., 2010*; *Dahlén et al., 2021*; *Li and Schooling, 2020*). The ABO system has also been associated with the susceptibility, progression, and severity of COVID-19 (*Ellinghaus et al., 2020*). In contrast, apart from hemolytic disease of the newborn, reported associations between the RhD blood group and disease development are sparser (*Anstee, 2010*).

Specifically, higher levels of factor VIII (FVIII) and von Willebrand factor (vWF) observed in individuals with a non-O blood group have been suggested to affect the development of cardiovascular disease (*Jenkins and O'Donnell, 2006*; *Franchini and Lippi, 2015*). Additionally, blood group-related antigens have been suggested to be involved in the adhesion of trophoblast, inflammatory cells, and metastatic tumor cells to the endothelial cells of the vasculature (*Ravn and Dabelsteen, 2000*). The endothelial cells of the vasculature have also been suggested to contribute to the initiation and propagation of severe clinical manifestations of COVID-19 (*Teuwen et al., 2020*).

Recently, an associative disease-wide risk analysis of the ABO and RhD blood groups was conducted in a large Swedish cohort (*Dahlén et al., 2021*). The study generated further support for previous findings and suggested new associations. Here, we further uncover the relationship between the ABO and RhD blood groups and disease susceptibility using a Danish cohort of 482,914 patients. In contrast to previous studies, we use up to 41 years of follow-up data, and a disease categorization scheme specifically developed for disease-wide analysis called phecodes (*Wu et al., 2019*). Further, we determine the uniqueness of each individual ABO blood group as opposed to using blood group O as the reference.

We estimate incidence rate ratios of 1312 phecodes (diagnoses) for the ABO and RhD blood groups. Further, we determine associations between the ABO/RhD blood groups and the age at the first diagnosis to better disclose the temporal life course element of disease development.

## Methods

### Study design

This retrospective cohort study was based on the integration of the Danish National Patient Registry (DNPR) and data on ABO/RhD blood groups of hospitalized patients. We included Danish patients who had their ABO/RhD blood group determined in the Capital Region or Region Zealand (covering ~45% of the Danish population *Nordjylland, 2022*), between January 1, 2006, and April 10, 2018. A blood type determination is commonly done for patients who may require a blood transfusion during hospitalization for example, anemic patients and women in labor. In the inclusion period, approximately 90% of the population in the Capital Region and 97% of the Region Zealand population were of European ancestry (*Supplementary file 1*). The DNPR provided the International Classification of Diseases 8th and 10th revision (ICD-8 and ICD-10) diagnosis codes, dates of diagnosis, date of birth, date of potential emigration, and sex of patients, with records dating back to 1977.

Similar to a case-control study, the patients were included retrospectively. Here, selection into the study was based on an in-hospital ABO/RhD blood group determination. That is, the person-time and the entire disease history back to 1977 of patients hospitalized between 2006 and 2018 with known ABO/RhD blood groups were included retrospectively.

We defined diseased and non-diseased individuals using the phecode mapping from ICD-10 diagnosis codes (*Wu et al., 2019*). Before categorizing the assigned ICD diagnosis codes into phecodes, the ICD-8 codes were converted to ICD-10 codes (*Pedersen et al., 2023*). Further, referral diagnoses were excluded. Pregnancy- and perinatal diagnosis (ICD-10 chapters 15–16) assigned before or after age 10 were excluded, or, when possible, rightly assigned to the mother or newborn, respectively. The disease categories of injuries, poisonings, and symptoms were deemed unlikely to be associated with the blood groups and excluded from the analyses (phecode categories: 'injuries and poisonings', 'symptoms' and phecodes above 999). Only phecodes with at least 100 cases in the study sample

were included. The patients were followed from the entry in the DNPR to the date of death, emigration, the first event of the studied phecode, or end study period (April 10, 2018), whichever came first. Thus, follow-up was up to 41 years. The patients were allowed to contribute events and time at risk to multiple phecode analysis.

## Diagnosis-wide incidence rate ratios

We used a log-linear quasi-Poisson regression model to estimate incidence rate ratios (IRRs) of each phecode among individuals with blood groups A, B, AB, and O relative to the other blood groups, respectively (e.g. A vs. B, AB, and O) (*Dewey et al., 1995*; *Ver Hoef and Boveng, 2007*). Further, we compared individuals with positive RhD type relative to negative RhD type. The analyses of diseases developed by both males and females were adjusted for sex, while analyses of sex-restricted diseases (e.g. cervical cancer) only included a subgroup of individuals of the restricted sex. Sex-restricted diseases were pre-defined by the phecode terminology. Sex was adjusted for as prior studies have found sex differences in the incidence rates of multiple diseases (*Westergaard et al., 2019*). Further, we adjusted for the year of birth and attained age, both modeled using restricted cubic splines with five knots. Attained age was split into 1 year intervals and treated as a time-dependent covariate, thus allowing individuals to move between categories with time. Herewith, age was used as the underlying time scale. Further, an interaction between attained age and sex was modeled for non-sex-restricted analyses. Patients were excluded from the analysis if they were assigned the phecode under study at the start of the DNPR. For analysis of congenital phecodes (e.g. sickle cell disease), prevalence ratios were estimated instead of IRRs by using the cohort size as the offset (see *Supplementary file 2* for a list of the congenital phecodes). The analyses of ABO blood groups were adjusted for RhD type, and the RhD-analyses were adjusted for the ABO blood group. Adjustment for the birth year was done to control for societal changes and was used instead of the calendar period of diagnosis. The robust quasi-Poisson variance formula was used to control for over-dispersion (*Ver Hoef and Boveng, 2007*). We conducted a supplemental analysis using the same methodology but where blood group O was instead used as the reference to enable direct comparison and meta-analysis with previous studies.

## Age of first hospital diagnosis

We estimated differences in age of first phecode of individuals with blood group A, B, AB, and O relative to any other blood group, respectively. Similar analyses were done for RhD-positive individuals relative to RhD negative individuals. We used a linear regression model adjusted for sex and birth year (as a restricted cubic spline with five knots). Analysis of sex-restricted phecodes was not adjusted for sex. Individuals who were assigned the studied phecode at the start date of the DNPR were excluded as the age of diagnosis was uncertain. Further, congenital- and pregnancy-related phecodes were not included.

Statistical analyses were performed in R (version 3.6.2) using the survival and rms package. p-values were two-sided. p-values and confidence intervals were adjusted for multiple testing by the false discovery rate (FDR) approach, accounting for the number of performed tests (5 blood groups times 1312 phecodes; *Benjamini and Hochberg, 1995*; *Altman and Bland, 2011*). FDR adjusted p-values <0.05 were deemed statistically significant. The analysis pipeline was made in python (anaconda3/5.3.0) using snakemake for reproducibility (*Köster and Rahmann, 2012*). The analyses code is available through https://www.github.com/peterbruun/blood_type_study (copy archived at *Bruun-Rasmussen, 2023*). The manuscript complies with the STROBE reporting guidelines.

## Results

In total, 482,914patients (60.4%females) were included and 1312 phecodes (diagnosis codes) were examined (*Figure 1*, and *Table 1*). The median follow-up time for all phecode analyses was 17,555,322 person-years (Q1-Q3: 17,324,597–17,615,142). The cohort held a wide age distribution of patients born from 1901 to 2015 (*Table 1*, and *Supplementary file 3*). The ABO/RhD blood group distribution of the patients was similar to that of a previously summarized reference population of 2.2 million Danes (*Table 1*; *Barnkob et al., 2020*; *Banks, 2022*).

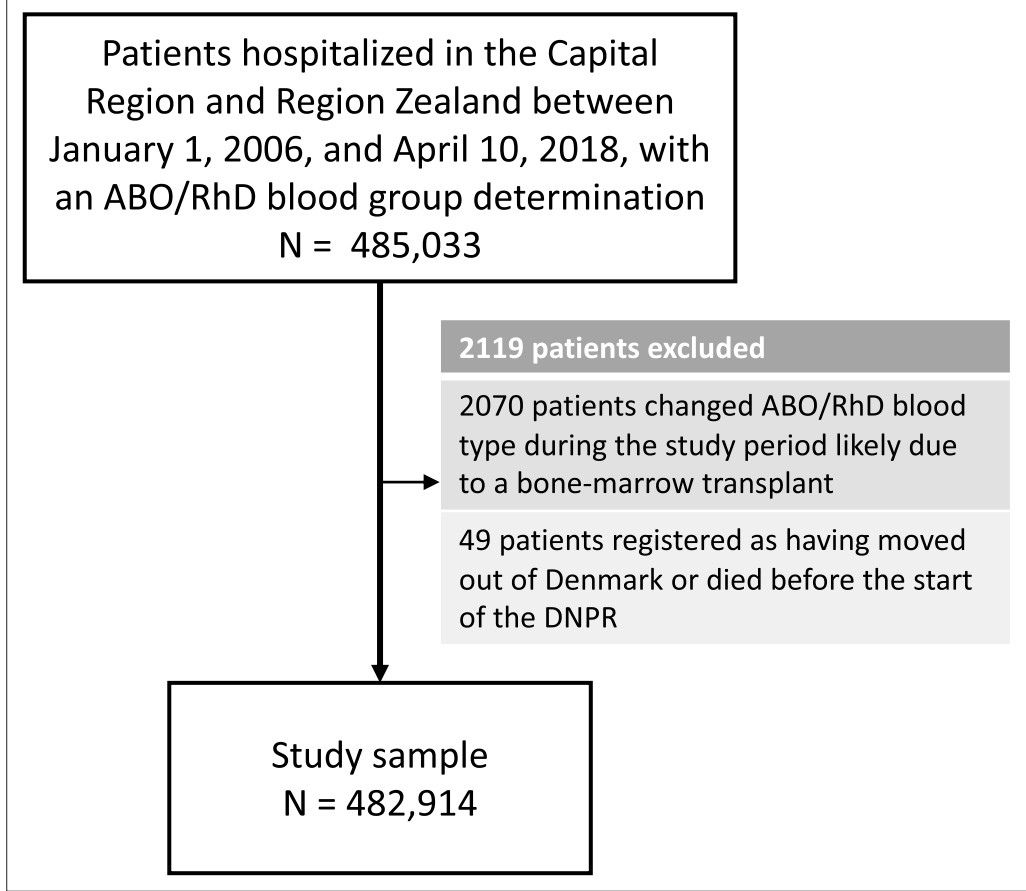

Patients hospitalized in the Capital Region and Region Zealand between January 1, 2006, and April 10, 2018, with an ABO/RhD blood group determination
N = 485,033

**2119 patients excluded**

2070 patients changed ABO/RhD blood type during the study period likely due to a bone-marrow transplant

49 patients registered as having moved out of Denmark or died before the start of the DNPR

Study sample
N = 482,914

**Figure 1.** Selection of patients for the 41-year retrospective cohort study on ABO/RhD blood groups and associations with disease incidence in 482,914 Danish patients.

**Table 1.** Characteristics of patients in the 41-year retrospective cohort study on ABO/RhD blood groups and associations with disease incidence in 482,914 Danish patients.

|  |  | N=482,914 |
|---|---|---|
|  | O | 197,634 (40.9) |
|  | A | 206,110 (42.7) |
|  | AB | 22,111 (4.6) |
| ABO, n (%) | B | 57,059 (11.8) |
|  | Negative | 74,150 (15.4) |
| RhD, n (%) | Positive | 408,764 (84.6) |
|  | K | 291,649 (60.4) |
| Sex, n (%) | M | 191,265 (39.6) |
| Birth year, median [Q1,Q3] |  | 1963 [1945,1982] |
| Age at entry, median [Q1,Q3] |  | 13 [0,32] |
| Follow-up time, median [Q1,Q3] |  | 40.8 [33.4,41.3] |

## Incidence rate ratios

After adjustment for multiple testing, we found the incidence rate ratios (IRRs) of 101 and 28 phecodes (116 unique) to be statistically significant for the ABO and RhD blood groups, respectively. The statistically significant IRRs are given with 95% confidence intervals in *Table 2*. The estimates of all examined phecodes are given in *Supplementary file 4*. Further, Manhattan plots of the p-values and disease categories are presented in *Figures 2–6* .

The number of statistically significant IRRs for A, B, AB, O, and RhD were 50, 38, 11, 53, and 28, respectively. However, a between blood group comparison on the number of statistically significant IRRs is problematic because the analyses of blood group A and O had the highest power given that these blood groups were most frequent in the study sample (*Table 1*). For 13 phecodes, an association was found for both the ABO blood group and the RhD blood group. The ABO blood groups were found positively associated with 75 phecodes and inversely associated

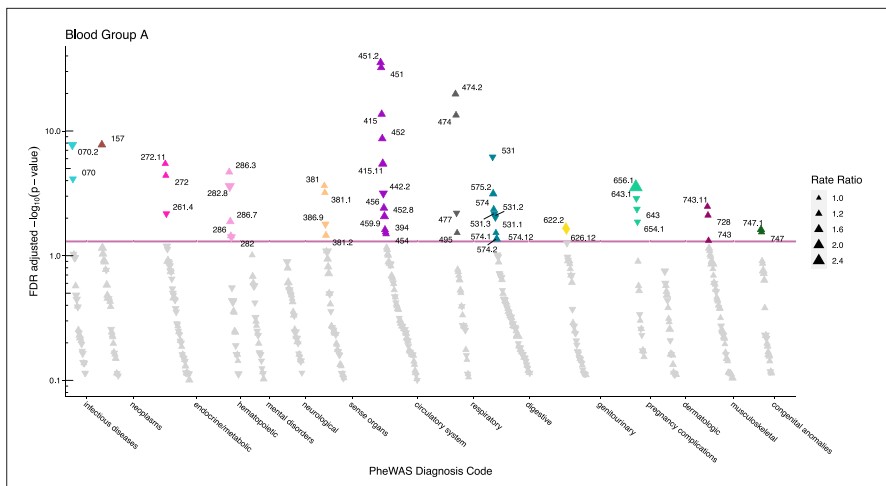

**Figure 2.** Manhattan plot for blood group A with phecodes included by category. The vertical axis shows the -log10 transformed FDR adjusted p-values on a log10-scale. The horizontal axis shows the phecodes by category. The red line indicates the statistically significant level of <0.05 for FDR adjusted p-values. Associations with p-value >0.8 are not displayed. Coloured and annotated associations were deemed statistically significant. The direction of the triangles indicates positive or inverse associations (upward: IRR >1, downward: IRR <1). The size of the triangles indicates the size of the incidence rate ratio.

with 67 phecodes. The RhD-positive blood group was found to have 16 positive- and 12 inverse associations. Blood groups A and O were associated with diseases of the circulatory and digestive system. Blood group B was associated with several infectious, metabolic, and musculoskeletal diseases. The associations of the RhD blood group included cancers, infectious diseases, and pregnancy complications. The results of the supplementary analyses where blood group O was used as the reference is shown in *Supplementary files 6 and 7*.

## Age at first diagnosis

We found the B blood group to be associated with a later diagnosis of viral infection. Further, blood group O was associated with a later diagnosis of phlebitis and thrombophlebitis (*Table 3* and

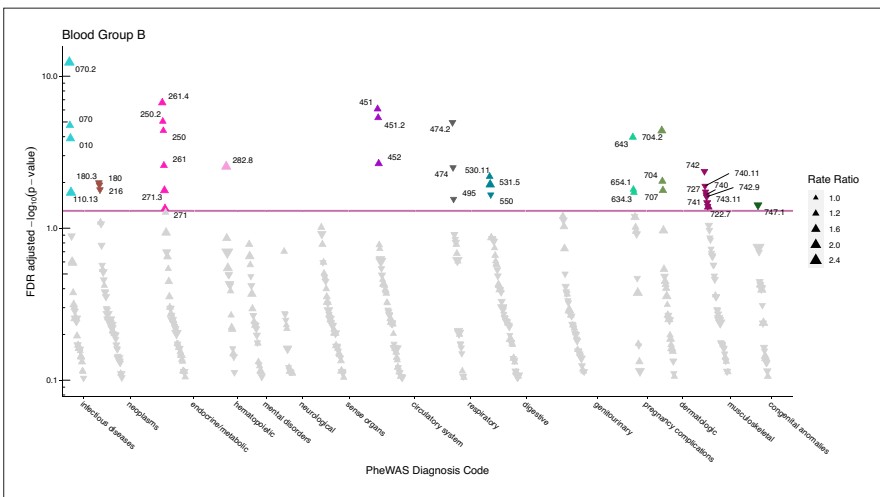

**Figure 3.** Manhattan plot for blood group B with phecodes included by category. The vertical axis shows the -log10 transformed FDR adjusted p-values on a log10-scale. The horizontal axis shows the phecodes by category. The red line indicates the statistically significant level of <0.05 for FDR adjusted p-values. Associations with p-value >0.8 are not displayed. Coloured and annotated associations were deemed statistically significant. The direction of the triangles indicates positive or inverse associations (upward: IRR >1, downward: IRR <1). The size of the triangles indicates the size of the incidence rate ratio.

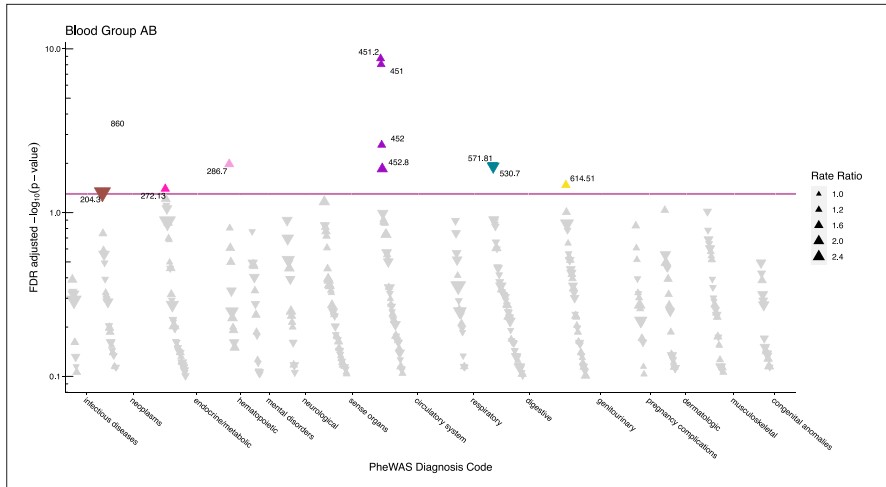

**Figure 4.** Manhattan plot for blood group AB with phecodes included by category. The vertical axis shows the -log10 transformed FDR adjusted p-values on a log10-scale. The horizontal axis shows the phecodes by category. The red line indicates the statistically significant level of <0.05 for FDR adjusted p-values. Associations with p-value >0.8 are not displayed. Coloured and annotated associations were deemed statistically significant. The direction of the triangles indicates positive or inverse associations (upward: IRR >1, downward: IRR <1). The size of the triangles indicates the size of the incidence rate ratio.

*Supplementary file 5*). The RhD-positive group was associated with a later diagnosis of acute and chronic tonsilitis diagnosis.

## Discussion

We found the ABO/RhD blood groups to be associated with a wide spectrum of diseases including cancers and musculoskeletal-, genitourinary-, endocrinal-, infectious-, cardiovascular-, and gastrointestinal diseases. Associations of the ABO blood groups included monocytic leukemia, tonsilitis, renal dialysis, diseases of the female reproductive system, and osteoarthrosis. Associations of the RhD blood group included cancer of the tongue, malignant neoplasm (other), tuberculosis-, HIV-, hepatitis

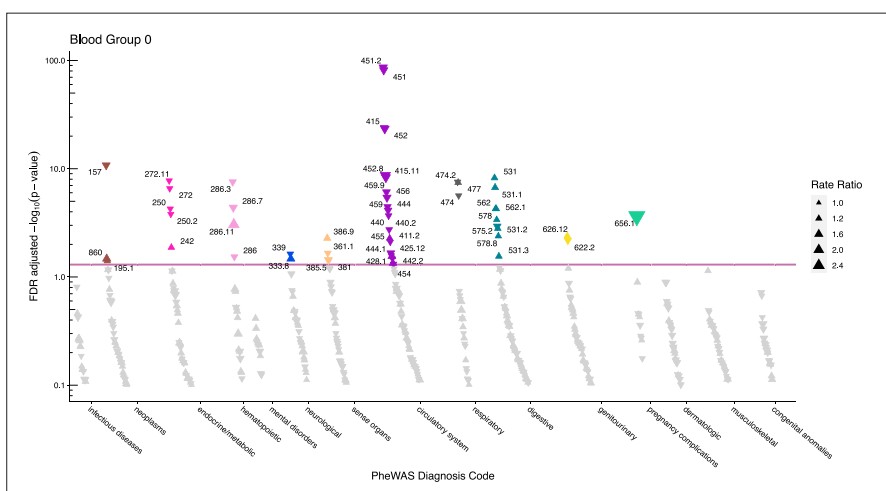

**Figure 5.** Manhattan plot for blood group O with phecodes included by category. The vertical axis shows the -log10 transformed FDR adjusted p-values on a log10-scale. The horizontal axis shows the phecodes by category. The red line indicates the statistically significant level of <0.05 for FDR adjusted p-values. Associations with p-value >0.8 are not displayed. Coloured and annotated associations were deemed statistically significant. The direction of the triangles indicates positive or inverse associations (upward: IRR >1, downward: IRR <1). The size of the triangles indicates the size of the incidence rate ratio.

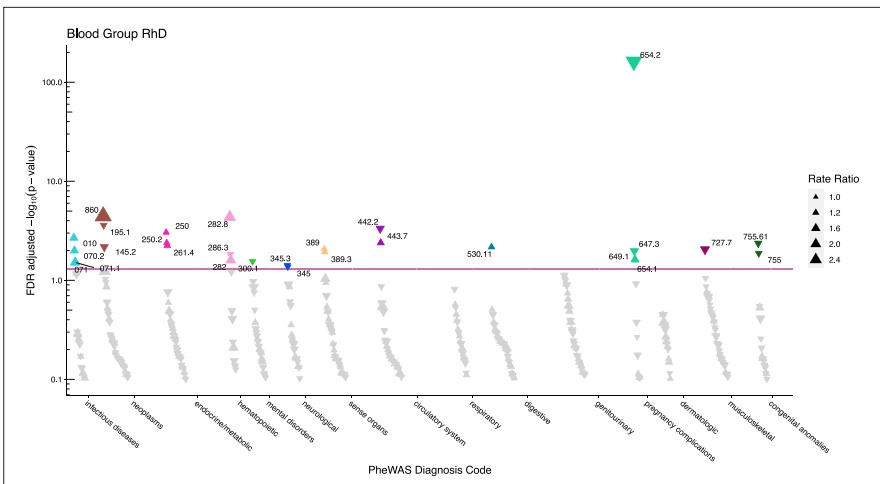

**Figure 6.** Manhattan plot for the Rhesus D blood group with phecodes included by category. The vertical axis shows the -log10 transformed FDR adjusted p-values on a log10-scale. The horizontal axis shows the phecodes by category. The red line indicates the statistically significant level of <0.05 for FDR adjusted p-values. Associations with p-value >0.8 are not displayed. Coloured and annotated associations were deemed statistically significant. The direction of the triangles indicates positive or inverse associations (upward: IRR >1, downward: IRR <1). The size of the triangles indicates the size of the incidence rate ratio.

B infection, type 2 diabetes, hereditary hemolytic anemias, major puerperal infection, anxiety disorders, and contracture of tendon.

The blood groups may reflect their corresponding genetic markers; thus, our findings may indicate an association between disease and the ABO locus on chromosome 9 and the RH locus on chromosome 1, respectively. Alternatively, the associations may indicate that the blood groups are involved in disease mechanisms at the molecular level mediated either through the blood group antigens or by the blood group reactive antibodies. However, our findings have a compromised causal interpretation given the retrospective inclusion of individuals (and person-time) after an in-hospital blood group test.

Our results support several previously observed associations including positive associations between the non-O blood groups and prothrombotic diseases of the circulatory system (phecodes: 411.1–459.9), associations with gastroduodenal ulcers, associations of blood group O and lower risk of type 2 diabetes, and positive association between blood group B and tuberculosis (*Vasan et al., 2016*; *Edgren et al., 2010*; *Dahlén et al., 2021*; *Fagherazzi et al., 2015*; *Rao, 2012*). Further, our results support findings associating non-O blood groups with increased risk of pancreatic cancer (*Liumbruno and Franchini, 2014*). The role of the ABO blood group in HIV susceptibility remains controversial; we only observed a positive association for the RhD-positive blood group (*Davison et al., 2020*).

We found blood group B to be positively associated with 'ectopic pregnancy', 'excessive vomiting in pregnancy, and 'abnormality of organs and soft tissues of pelvis complicating pregnancy' indicating that blood group B mothers may be more likely to experience pregnancy complications. Further, we found positive associations of blood group A with both 'mucous polyp of cervix', and blood group AB with 'cervicitis and endocervicitis'. Taken together these findings may indicate that the ABO blood groups are associated with diseases of the female reproductive system. However, the study design does not allow for any causal interpretation.

Only a few statistically significant associations were found for the analyses of the age of the first diagnosis; thus, indicating that the blood group's involvement in disease onset may be marginal. However, we assumed a linear relationship with age because assessing potential non-linear relationships for each disease would be unfeasible given the large number of tests performed. The linearity assumption may not hold for all analyses which limits the interpretation of the estimates.

A strength of our approach is that we utilized the phecode disease classification scheme that is specifically developed for disease-wide risk analyses (*Wu et al., 2019*) The phecode mapping scheme combines ICD-10 codes that clinical domain experts have deemed to cover the same disease. For

**Table 2.** Statistically significant incidence rate ratios for each individual blood group A, B, AB, and O relative to any other blood group (e.g. A vs. B, AB, and O combined). Further, also for RhD-positive blood group relative to the RhD negative blood group.

| Phecode | Phenotype | Cases | Person-years | Blood group A IRR (95% CI) | p-value | Blood group B IRR (95% CI) | p-value | Blood group AB IRR (95% CI) | p-value | Blood group 0 IRR (95% CI) | p-value | Blood group RhD IRR (95% CI) | p-value |
|---|---|---|---|---|---|---|---|---|---|---|---|---|---|
| **Infectious Diseases** | | | | | | | | | | | | | |
| 010 | Tuberculosis | 2101 | 17603440 | 0.87 (0.74, 1.02) | 0.093 | **1.4 (1.18, 1.65)** | **<0.001** | 1.07 (0.32, 3.57) | 0.915 | 0.97 (0.59, 1.57) | 0.899 | **1.36 (1.12, 1.65)** | **0.002** |
| 070 | Viral hepatitis | 6596 | 17557078 | **0.88 (0.82, 0.93)** | **<0.001** | **1.22 (1.12, 1.34)** | **<0.001** | 1 (0.81, 1.23) | 0.97 | 1.04 (0.9, 1.21) | 0.583 | 1.12 (0.99, 1.28) | 0.069 |
| 070.2 | Viral hepatitis B | 1664 | 17613572 | **0.7 (0.62, 0.79)** | **<0.001** | **1.66 (1.46, 1.9)** | **<0.001** | 1.12 (0.38, 3.26) | 0.851 | 1.06 (0.7, 1.61) | 0.794 | **1.36 (1.07, 1.71)** | **0.01** |
| 071 | Human immunodeficiency virus [HIV] disease | 1182 | 17620808 | 0.81 (0.63, 1.05) | 0.109 | 1.18 (0.54, 2.59) | 0.688 | 1.19 (0.52, 2.73) | 0.693 | 1.1 (0.67, 1.81) | 0.725 | **1.49 (1.04, 2.15)** | **0.031** |
| 071.1 | HIV infection, symptomatic | 1182 | 17620808 | 0.81 (0.63, 1.05) | 0.109 | 1.18 (0.54, 2.59) | 0.688 | 1.19 (0.52, 2.73) | 0.693 | 1.1 (0.67, 1.81) | 0.725 | **1.49 (1.04, 2.15)** | **0.031** |
| 110.13 | Dermatophytosis of the body | 211 | 17630531 | 0.89 (0.35, 2.22) | 0.809 | **1.64 (1.08, 2.49)** | **0.02** | 0.93 (0.03, 30.15) | 0.97 | 0.88 (0.35, 2.23) | 0.804 | 1.13 (0.2, 6.46) | 0.897 |
| **Neoplasms** | | | | | | | | | | | | | |
| 145.2 | Cancer of tongue | 606 | 17629055 | 0.95 (0.51, 1.76) | 0.877 | 1.13 (0.71, 1.82) | 0.616 | 1.21 (0.43, 3.38) | 0.726 | 0.96 (0.46, 2.03) | 0.93 | **0.74 (0.6, 0.92)** | **0.007** |
| 157 | Pancreatic cancer | 2828 | 17627948 | **1.26 (1.16, 1.36)** | **<0.001** | 1.02 (0.34, 3.13) | 0.97 | 1.11 (0.53, 2.33) | 0.8 | **0.76 (0.7, 0.82)** | **<0.001** | 1.01 (0.52, 1.98) | 0.97 |
| 180 | Cervical cancer and dysplasia | 12538 | 10308860 | 1.05 (0.99, 1.12) | 0.118 | **0.9 (0.83, 0.97)** | **0.01** | 0.96 (0.66, 1.39) | 0.839 | 1 (0.99, 1.01) | 0.97 | 0.93 (0.86, 1.01) | 0.083 |
| 180.3 | Cervical intraepithelial neoplasia [CIN] [Cervical dysplasia] | 10895 | 10327107 | 1.06 (0.99, 1.13) | 0.115 | **0.89 (0.82, 0.97)** | **0.009** | 0.95 (0.67, 1.34) | 0.766 | 1 (0.84, 1.19) | 0.97 | 0.93 (0.85, 1.01) | 0.102 |
| 195.1 | Malignant neoplasm, other | 7383 | 17603424 | 0.96 (0.87, 1.05) | 0.368 | 0.95 (0.77, 1.17) | 0.642 | 0.96 (0.55, 1.69) | 0.906 | **1.07 (1, 1.15)** | **0.038** | **0.88 (0.82, 0.94)** | **<0.001** |
| 204.3 | Monocytic leukemia | 179 | 17631300 | 1.04 (0.15, 7.27) | 0.97 | 0.92 (0.06, 13.67) | 0.957 | **0.25 (0.06, 0.98)** | **0.047** | 1.13 (0.44, 2.91) | 0.809 | 0.84 (0.37, 1.93) | 0.698 |
| 216 | Benign neoplasm of skin | 12993 | 17495431 | 1.02 (0.84, 1.23) | 0.883 | **0.9 (0.82, 0.98)** | **0.014** | 0.99 (0.62, 1.57) | 0.97 | 1.03 (0.93, 1.14) | 0.583 | 0.95 (0.85, 1.05) | 0.32 |
| 860 | Bone marrow or stem cell transplant | 142 | 17631302 | 1.01 (0.77, 1.31) | 0.97 | 0.63 (0.15, 2.57) | 0.531 | **0.15 (0.05, 0.44)** | **<0.001** | **1.37 (1.02, 1.82)** | **0.033** | **4.15 (2.13, 8.09)** | **<0.001** |
| **Endocrine/Metabolic** | | | | | | | | | | | | | |
| 242 | Thyrotoxicosis with or without goiter | 9744 | 17527426 | 0.93 (0.86, 1.02) | 0.127 | 0.99 (0.59, 1.65) | 0.97 | 0.9 (0.66, 1.22) | 0.505 | **1.09 (1.02, 1.18)** | **0.013** | 1.02 (0.7, 1.48) | 0.927 |
| 250 | Diabetes mellitus | 36810 | 17295033 | 1.01 (0.92, 1.11) | 0.849 | **1.09 (1.05, 1.13)** | **<0.001** | 1.06 (0.95, 1.18) | 0.32 | **0.95 (0.92, 0.97)** | **<0.001** | **1.07 (1.03, 1.11)** | **<0.001** |
| 250.2 | Type 2 diabetes | 32505 | 17346533 | 1 (0.82, 1.23) | 0.97 | **1.1 (1.06, 1.15)** | **<0.001** | 1.07 (0.96, 1.19) | 0.213 | **0.94 (0.92, 0.97)** | **<0.001** | **1.07 (1.02, 1.12)** | **0.004** |
| 261 | Vitamin deficiency | 6674 | 17594082 | 0.94 (0.86, 1.03) | 0.183 | **1.15 (1.05, 1.26)** | **0.003** | 0.93 (0.66, 1.31) | 0.683 | 1.01 (0.75, 1.36) | 0.94 | 1.06 (0.92, 1.22) | 0.405 |
| 261.4 | Vitamin D deficiency | 4105 | 17613013 | **0.91 (0.85, 0.97)** | **0.007** | **1.23 (1.14, 1.34)** | **<0.001** | 0.95 (0.56, 1.61) | 0.86 | 1.01 (0.63, 1.63) | 0.97 | **1.14 (1.04, 1.25)** | **0.006** |
| 271 | Disorders of carbohydrate transport and metabolism | 1306 | 17619038 | 0.86 (0.69, 1.07) | 0.176 | **1.28 (1, 1.64)** | **0.047** | 1.07 (0.23, 4.92) | 0.934 | 1.01 (0.51, 2) | 0.97 | 1.16 (0.78, 1.72) | 0.475 |
| 271.3 | Intestinal disaccharidase deficiencies and disaccharide malabsorption | 1154 | 17621337 | 0.85 (0.68, 1.05) | 0.135 | **1.32 (1.05, 1.67)** | **0.018** | 1.08 (0.24, 4.88) | 0.924 | 1.01 (0.55, 1.86) | 0.97 | 1.15 (0.73, 1.8) | 0.562 |
| 272 | Disorders of lipoid metabolism | 41222 | 17347032 | **1.06 (1.03, 1.09)** | **<0.001** | 1.01 (0.72, 1.42) | 0.97 | 1.03 (0.63, 1.69) | 0.92 | **0.93 (0.91, 0.96)** | **<0.001** | 1.02 (0.9, 1.17) | 0.733 |
| 272.11 | Hypercholesterolemia | 35565 | 17395012 | **1.06 (1.03, 1.09)** | **<0.001** | 1.01 (0.79, 1.29) | 0.956 | 1.03 (0.88, 1.21) | 0.709 | **0.93 (0.91, 0.96)** | **<0.001** | 1.03 (0.96, 1.09) | 0.433 |
| 272.13 | Mixed hyperlipidemia | 1324 | 17619911 | 1.05 (0.66, 1.67) | 0.841 | 1.04 (0.14, 7.94) | 0.97 | **1.39 (1.01, 1.91)** | **0.04** | 0.87 (0.7, 1.08) | 0.211 | 1.09 (0.58, 2.04) | 0.809 |
| **Hematopoietic** | | | | | | | | | | | | | |

*Table 2 continued on next page*

*Table 2 continued*

| Phecode | Phenotype | Cases | Person-years | Blood group A IRR (95% CI) | p-value | Blood group B IRR (95% CI) | p-value | Blood group AB IRR (95% CI) | p-value | Blood group 0 IRR (95% CI) | p-value | Blood group RhD IRR (95% CI) | p-value |
|---|---|---|---|---|---|---|---|---|---|---|---|---|---|
| 282 | Hereditary hemolytic anemias | 947 | 482914* | **0.76 (0.59, 0.99)**\*\* | 0.039 | 1.36 (0.9, 2.06)\*\* | 0.141 | 1.42 (0.72, 2.8)\*\* | 0.316 | 1.03 (0.25, 4.26)\*\* | 0.97 | **1.65 (1.06, 2.56)**\*\* | 0.026 |
| 282.8 | Other hemoglobinopathies | 557 | 482914* | **0.65 (0.51, 0.82)**\*\* | <0.001 | **1.56 (1.16, 2.1)**\*\* | 0.003 | 1.47 (0.77, 2.83)\*\* | 0.248 | 1.09 (0.58, 2.07)\*\* | 0.798 | **2.24 (1.52, 3.29)**\*\* | <0.001 |
| 286 | Coagulation defects | 4124 | 17606796 | **1.14 (1.01, 1.3)** | 0.035 | 0.95 (0.56, 1.6) | 0.854 | 1.14 (0.67, 1.95) | 0.633 | **0.87 (0.76, 0.99)** | 0.029 | 0.93 (0.66, 1.33) | 0.717 |
| 286.11 | Von willebrand's disease | 214 | 482914* | 0.71 (0.35, 1.47)\*\* | 0.368 | 0.49 (0.17, 1.45)\*\* | 0.199 | 0.39 (0.02, 8.74)\*\* | 0.562 | **1.96 (1.32, 2.91)**\*\* | <0.001 | 0.68 (0.29, 1.6)\*\* | 0.388 |
| 286.3 | Coagulation defects complicating pregnancy or postpartum | 2015 | 10502418 | **1.15 (1.08, 1.22)** | <0.001 | 1.02 (0.58, 1.79) | 0.946 | 1.14 (0.95, 1.38) | 0.164 | **0.84 (0.79, 0.89)** | <0.001 | **0.88 (0.8, 0.97)** | 0.013 |
| 286.7 | Other and unspecified coagulation defects | 1085 | 17621421 | **1.23 (1.05, 1.45)** | 0.013 | 0.98 (0.45, 2.16) | 0.97 | **1.52 (1.1, 2.08)** | 0.011 | **0.74 (0.64, 0.85)** | <0.001 | 1.03 (0.23, 4.72) | 0.97 |
| **Mental Disorders** | | | | | | | | | | | | | |
| 300.1 | Anxiety disorder | 7985 | 17603188 | 1.03 (0.93, 1.14) | 0.575 | 0.97 (0.8, 1.16) | 0.735 | 0.96 (0.66, 1.41) | 0.858 | 0.99 (0.76, 1.29) | 0.953 | **0.92 (0.86, 0.99)** | 0.027 |
| **Neurological** | | | | | | | | | | | | | |
| 333.8 | Other degenerative diseases of the basal ganglia | 381 | 17630209 | 0.78 (0.57, 1.07) | 0.128 | 1.05 (0.11, 9.95) | 0.967 | 0.65 (0.24, 1.78) | 0.407 | **1.32 (1.02, 1.71)** | 0.033 | 1 (0.95, 1.05) | 0.97 |
| 339 | Other headache syndromes | 3466 | 17603269 | 1.05 (0.9, 1.22) | 0.546 | 1.11 (0.95, 1.3) | 0.204 | 1.03 (0.39, 2.73) | 0.957 | **0.9 (0.83, 0.99)** | 0.023 | 1.06 (0.82, 1.36) | 0.688 |
| 345 | Epilepsy, recurrent seizures, convulsions | 23469 | 17228780 | 0.99 (0.77, 1.28) | 0.97 | 0.96 (0.86, 1.08) | 0.51 | 0.97 (0.71, 1.32) | 0.853 | 1.03 (0.96, 1.1) | 0.44 | **0.94 (0.89, 1)** | 0.039 |
| 345.3 | Convulsions | 14391 | 17351676 | 0.99 (0.76, 1.3) | 0.97 | 0.96 (0.79, 1.18) | 0.732 | 1 (0.88, 1.13) | 0.97 | 1.02 (0.89, 1.18) | 0.77 | **0.92 (0.85, 0.99)** | 0.036 |
| **Sense Organs** | | | | | | | | | | | | | |
| 361.1 | Retinal detachment with retinal defect | 3037 | 17594394 | 1.06 (0.87, 1.29) | 0.582 | 1.07 (0.69, 1.66) | 0.787 | 1.23 (0.9, 1.69) | 0.193 | **0.88 (0.79, 0.98)** | 0.022 | 0.97 (0.49, 1.96) | 0.948 |
| 381 | Otitis media and Eustachian tube disorders | 22790 | 17144551 | **1.07 (1.03, 1.11)** | <0.001 | 0.94 (0.86, 1.04) | 0.226 | 1.01 (0.55, 1.87) | 0.97 | **0.95 (0.91, 1)** | 0.039 | 0.96 (0.87, 1.06) | 0.437 |
| 381.1 | Otitis media | 12313 | 17364091 | **1.09 (1.04, 1.14)** | <0.001 | 0.95 (0.83, 1.09) | 0.449 | 0.98 (0.44, 2.22) | 0.97 | 0.94 (0.89, 1) | 0.067 | 0.97 (0.82, 1.13) | 0.688 |
| 381.2 | Eustachian tube disorders | 2358 | 17571118 | **1.15 (1.01, 1.31)** | 0.033 | 0.9 (0.62, 1.29) | 0.566 | 0.87 (0.38, 2.01) | 0.765 | 0.93 (0.74, 1.17) | 0.531 | 0.88 (0.69, 1.13) | 0.32 |
| 385.5 | Tympanosclerosis and middle ear disease related to otitis media | 530 | 17625215 | 1.25 (0.95, 1.64) | 0.114 | 0.99 (0.55, 1.77) | 0.97 | 1.21 (0.37, 3.96) | 0.761 | **0.77 (0.6, 0.98)** | 0.036 | 1.08 (0.29, 3.98) | 0.918 |
| 386.9 | Dizziness and giddiness (Light-headedness and vertigo) | 1060 | 17624097 | **0.84 (0.73, 0.97)** | 0.016 | 1.04 (0.43, 2.55) | 0.933 | 0.82 (0.42, 1.57) | 0.555 | **1.2 (1.06, 1.37)** | 0.005 | 1.08 (0.7, 1.68) | 0.738 |
| 389 | Hearing loss | 43238 | 17166114 | 0.99 (0.88, 1.12) | 0.897 | 1.01 (0.74, 1.37) | 0.97 | 1.01 (0.78, 1.3) | 0.97 | 1.01 (0.83, 1.21) | 0.96 | **1.06 (1.01, 1.1)** | 0.009 |
| 389.3 | Degenerative and vascular disorders of ear | 22354 | 17419947 | 0.99 (0.81, 1.22) | 0.94 | 1 (0.85, 1.19) | 0.97 | 1.02 (0.53, 1.94) | 0.961 | 1 (0.83, 1.22) | 0.97 | **1.08 (1.02, 1.15)** | 0.012 |
| **Circulatory System** | | | | | | | | | | | | | |
| 394 | Rheumatic disease of the heart valves | 8422 | 17577658 | **1.08 (1.01, 1.16)** | 0.029 | 0.98 (0.61, 1.57) | 0.924 | 0.93 (0.63, 1.38) | 0.744 | 0.94 (0.86, 1.03) | 0.2 | 0.96 (0.78, 1.19) | 0.729 |
| 411.2 | Myocardial infarction | 25905 | 17411193 | 1.03 (0.96, 1.11) | 0.434 | 1.06 (0.97, 1.16) | 0.188 | 1.03 (0.58, 1.8) | 0.932 | **0.94 (0.9, 0.98)** | 0.008 | 1.02 (0.83, 1.26) | 0.867 |
| 415 | Pulmonary heart disease | 10870 | 17565369 | **1.2 (1.15, 1.26)** | <0.001 | 1.07 (0.9, 1.27) | 0.475 | 1.14 (0.96, 1.36) | 0.125 | **0.78 (0.75, 0.82)** | <0.001 | 0.95 (0.82, 1.09) | 0.482 |
| 415.11 | Pulmonary embolism and infarction, acute | 1533 | 17612465 | **1.39 (1.21, 1.59)** | <0.001 | 1.11 (0.56, 2.18) | 0.781 | 1.14 (0.44, 2.98) | 0.798 | **0.65 (0.57, 0.75)** | <0.001 | 0.88 (0.59, 1.32) | 0.562 |
| 425.12 | Other hypertrophic cardiomyopathy | 466 | 17628820 | 0.81 (0.61, 1.07) | 0.138 | 0.83 (0.35, 1.98) | 0.683 | 1.11 (0.12, 10.23) | 0.93 | **1.3 (1.03, 1.64)** | 0.028 | 0.91 (0.32, 2.6) | 0.877 |
| 428.1 | Congestive heart failure (CHF) NOS | 8357 | 17595328 | 1.03 (0.94, 1.13) | 0.5 | 1.02 (0.76, 1.37) | 0.906 | 1.12 (0.96, 1.3) | 0.14 | **0.94 (0.89, 0.99)** | 0.028 | 1.01 (0.7, 1.46) | 0.961 |

*Table 2 continued on next page*

*Table 2 continued*

| Phecode | Phenotype | Cases | Person-years | Blood group A IRR (95% CI) | p-value | Blood group B IRR (95% CI) | p-value | Blood group AB IRR (95% CI) | p-value | Blood group 0 IRR (95% CI) | p-value | Blood group RhD IRR (95% CI) | p-value |
|---|---|---|---|---|---|---|---|---|---|---|---|---|---|
| 440 | Atherosclerosis | 10901 | 17554704 | 1.05 (0.95, 1.16) | 0.345 | 1.1 (0.96, 1.25) | 0.169 | 1.12 (0.91, 1.37) | 0.3 | 0.9 (0.85, 0.95) | <0.001 | 1.04 (0.88, 1.23) | 0.675 |
| 440.2 | Atherosclerosis of the extremities | 8348 | 17570336 | 1.06 (0.96, 1.18) | 0.258 | 1.09 (0.9, 1.32) | 0.406 | 1.08 (0.75, 1.55) | 0.686 | 0.9 (0.84, 0.96) | 0.002 | 1.03 (0.8, 1.33) | 0.823 |
| 442.2 | Aneurysm of iliac artery | 326 | 17630660 | 0.75 (0.64, 0.89) | <0.001 | 1.13 (0.63, 2.04) | 0.69 | 1.2 (0.61, 2.38) | 0.607 | 1.21 (1, 1.47) | 0.045 | 0.72 (0.6, 0.87) | <0.001 |
| 443 | Peripheral vascular disease | 13791 | 17546796 | 1.05 (0.98, 1.12) | 0.153 | 1.01 (0.59, 1.73) | 0.97 | 1.03 (0.71, 1.5) | 0.872 | 0.94 (0.89, 1) | 0.035 | 1.03 (0.91, 1.18) | 0.63 |
| 443.7 | Peripheral angiopathy in diseases classified elsewhere | 3677 | 17611414 | 1.02 (0.68, 1.51) | 0.94 | 1.1 (0.9, 1.34) | 0.37 | 1.1 (0.78, 1.55) | 0.603 | 0.93 (0.83, 1.04) | 0.214 | 1.18 (1.05, 1.31) | 0.004 |
| 444 | Arterial embolism and thrombosis | 2390 | 17614619 | 1.14 (0.98, 1.34) | 0.093 | 1.11 (0.77, 1.61) | 0.583 | 1.28 (0.93, 1.76) | 0.134 | 0.79 (0.7, 0.89) | <0.001 | 1 (0.92, 1.08) | 0.97 |
| 444.1 | Arterial embolism and thrombosis of lower extremity artery | 1286 | 17623872 | 1.13 (0.78, 1.63) | 0.533 | 1.13 (0.61, 2.13) | 0.706 | 1.38 (0.86, 2.23) | 0.185 | 0.78 (0.63, 0.97) | 0.022 | 0.98 (0.35, 2.76) | 0.97 |
| 451 | Phlebitis and thrombophlebitis | 16748 | 17479709 | 1.22 (1.18, 1.25) | <0.001 | 1.15 (1.09, 1.21) | <0.001 | 1.29 (1.18, 1.4) | <0.001 | 0.73 (0.7, 0.75) | <0.001 | 0.97 (0.84, 1.13) | 0.717 |
| 451.2 | Phlebitis and thrombophlebitis of lower extremities | 15650 | 17489528 | 1.23 (1.19, 1.27) | <0.001 | 1.14 (1.08, 1.2) | <0.001 | 1.31 (1.2, 1.42) | <0.001 | 0.72 (0.69, 0.74) | <0.001 | 0.97 (0.84, 1.12) | 0.678 |
| 452 | Other venous embolism and thrombosis | 4275 | 17607194 | 1.24 (1.16, 1.32) | <0.001 | 1.21 (1.07, 1.36) | 0.002 | 1.31 (1.1, 1.56) | 0.003 | 0.69 (0.64, 0.74) | <0.001 | 1 (0.89, 1.11) | 0.97 |
| 452.8 | Postphlebitic syndrome | 341 | 17629425 | 1.35 (1.08, 1.68) | 0.009 | 1.26 (0.76, 2.08) | 0.378 | 1.82 (1.13, 2.94) | 0.014 | 0.55 (0.45, 0.67) | <0.001 | 1.18 (0.54, 2.56) | 0.688 |
| 454 | Varicose veins | 16500 | 17381971 | 1.1 (1.01, 1.2) | 0.032 | 0.98 (0.57, 1.68) | 0.946 | 1.04 (0.51, 2.1) | 0.93 | 0.91 (0.83, 1) | 0.05 | 0.98 (0.67, 1.42) | 0.906 |
| 455 | Hemorrhoids | 9001 | 17523962 | 0.95 (0.84, 1.08) | 0.481 | 0.91 (0.78, 1.08) | 0.279 | 0.91 (0.67, 1.24) | 0.562 | 1.1 (1.03, 1.18) | 0.005 | 0.99 (0.67, 1.46) | 0.97 |
| 456 | Chronic venous insufficiency [CVI] | 925 | 17626709 | 1.24 (1.07, 1.44) | 0.004 | 1.04 (0.23, 4.7) | 0.967 | 1.37 (0.91, 2.05) | 0.131 | 0.73 (0.64, 0.83) | <0.001 | 0.94 (0.49, 1.79) | 0.858 |
| 459 | Other disorders of circulatory system | 2555 | 17616713 | 1.12 (0.99, 1.27) | 0.076 | 1.12 (0.85, 1.48) | 0.433 | 1.15 (0.75, 1.75) | 0.528 | 0.82 (0.75, 0.9) | <0.001 | 1.08 (0.8, 1.46) | 0.636 |
| 459.9 | Circulatory disease NEC | 2174 | 17618930 | 1.16 (1.02, 1.31) | 0.024 | 1.13 (0.86, 1.5) | 0.381 | 1.22 (0.85, 1.74) | 0.279 | 0.78 (0.71, 0.86) | <0.001 | 1.02 (0.38, 2.74) | 0.966 |
| **Respiratory** | | | | | | | | | | | | | |
| 474 | Acute and chronic tonsillitis | 41427 | 16817602 | 1.1 (1.08, 1.13) | <0.001 | 0.93 (0.88, 0.97) | 0.002 | 1.01 (0.63, 1.63) | 0.97 | 0.94 (0.91, 0.96) | <0.001 | 0.97 (0.9, 1.05) | 0.466 |
| 474.2 | Chronic tonsillitis and adenoiditis | 27077 | 17030480 | 1.14 (1.11, 1.17) | <0.001 | 0.89 (0.84, 0.93) | <0.001 | 1.02 (0.75, 1.4) | 0.897 | 0.92 (0.89, 0.94) | <0.001 | 0.96 (0.89, 1.03) | 0.259 |
| 477 | Epistaxis or throat hemorrhage | 12337 | 17506794 | 0.93 (0.89, 0.98) | 0.006 | 0.92 (0.83, 1.02) | 0.118 | 0.94 (0.71, 1.23) | 0.648 | 1.12 (1.08, 1.16) | <0.001 | 1.01 (0.73, 1.41) | 0.948 |
| 495 | Asthma | 31106 | 17238738 | 1.04 (1, 1.08) | 0.028 | 0.94 (0.88, 0.99) | 0.023 | 1 (1, 1) | 0.97 | 0.99 (0.9, 1.08) | 0.777 | 0.99 (0.78, 1.27) | 0.97 |
| **Digestive** | | | | | | | | | | | | | |
| 530.11 | GERD | 7461 | 17582158 | 0.99 (0.69, 1.43) | 0.966 | 1.12 (1.03, 1.22) | 0.007 | 1.04 (0.62, 1.72) | 0.897 | 0.95 (0.88, 1.03) | 0.223 | 1.12 (1.03, 1.21) | 0.007 |
| 530.7 | Gastroesophageal laceration-hemorrhage syndrome | 597 | 17623900 | 1.16 (0.8, 1.66) | 0.439 | 1.04 (0.15, 7.23) | 0.97 | 0.43 (0.22, 0.83) | 0.012 | 0.94 (0.44, 2.02) | 0.883 | 1.09 (0.39, 2.99) | 0.883 |
| 531 | Peptic ulcer (excl. esophageal) | 16678 | 17443704 | 0.9 (0.87, 0.94) | <0.001 | 1 (0.94, 1.07) | 0.97 | 0.92 (0.75, 1.14) | 0.471 | 1.12 (1.08, 1.16) | <0.001 | 1.02 (0.83, 1.25) | 0.86 |
| 531.1 | Hemorrhage from gastrointestinal ulcer | 6277 | 17583751 | 0.91 (0.84, 0.98) | 0.009 | 0.89 (0.75, 1.04) | 0.149 | 0.91 (0.6, 1.4) | 0.688 | 1.17 (1.11, 1.25) | <0.001 | 1.07 (0.91, 1.27) | 0.421 |
| 531.2 | Gastric ulcer | 6745 | 17555312 | 0.91 (0.85, 0.98) | 0.008 | 1.01 (0.63, 1.63) | 0.97 | 0.92 (0.62, 1.36) | 0.681 | 1.11 (1.04, 1.18) | 0.002 | 0.97 (0.73, 1.28) | 0.838 |
| 531.3 | Duodenal ulcer | 4534 | 17555684 | 0.88 (0.8, 0.96) | 0.007 | 1.06 (0.71, 1.57) | 0.803 | 0.97 (0.21, 4.56) | 0.97 | 1.12 (1.01, 1.24) | 0.028 | 1.05 (0.77, 1.43) | 0.789 |
| 531.5 | Gastrojejunal ulcer | 586 | 17628539 | 0.83 (0.61, 1.11) | 0.213 | 1.41 (1.08, 1.84) | 0.012 | 1.12 (0.23, 5.41) | 0.897 | 1.01 (0.64, 1.6) | 0.97 | 0.85 (0.56, 1.28) | 0.439 |

*Table 2 continued on next page*

Table 2 continued

| Phecode | Phenotype | Cases | Person-years | Blood group A IRR (95% CI) | p-value | Blood group B IRR (95% CI) | p-value | Blood group AB IRR (95% CI) | p-value | Blood group 0 IRR (95% CI) | p-value | Blood group RhD IRR (95% CI) | p-value |
|---|---|---|---|---|---|---|---|---|---|---|---|---|---|
| 550 | Abdominal hernia | 47761 | 16976073 | 1.01 (0.94, 1.09) | 0.757 | **0.94 (0.89, 0.99)** | **0.021** | 0.97 (0.81, 1.17) | 0.765 | 1.02 (0.96, 1.08) | 0.575 | 1.01 (0.77, 1.32) | 0.97 |
| 562 | Diverticulosis and diverticulitis | 16569 | 17515568 | 0.94 (0.87, 1.03) | 0.19 | 0.92 (0.82, 1.03) | 0.138 | 0.91 (0.7, 1.18) | 0.481 | **1.11 (1.06, 1.17)** | **<0.001** | 0.98 (0.75, 1.28) | 0.884 |
| 562.1 | Diverticulosis | 16569 | 17515568 | 0.94 (0.87, 1.03) | 0.19 | 0.92 (0.82, 1.03) | 0.138 | 0.91 (0.7, 1.18) | 0.481 | **1.11 (1.06, 1.17)** | **<0.001** | 0.98 (0.75, 1.28) | 0.884 |
| 571.81 | Portal hypertension | 1101 | 17627608 | 0.94 (0.66, 1.35) | 0.766 | 1.17 (0.83, 1.63) | 0.376 | **0.62 (0.43, 0.9)** | **0.011** | 1.06 (0.76, 1.47) | 0.746 | 0.99 (0.52, 1.88) | 0.97 |
| 574 | Cholelithiasis and cholecystitis | 31530 | 17299206 | **1.05 (1.02, 1.09)** | **0.004** | 0.98 (0.85, 1.13) | 0.766 | 0.99 (0.64, 1.53) | 0.97 | 0.96 (0.92, 1) | 0.062 | 1.01 (0.73, 1.39) | 0.957 |
| 574.1 | Cholelithiasis | 24028 | 17364917 | **1.05 (1.01, 1.1)** | **0.029** | 0.98 (0.81, 1.18) | 0.809 | 0.99 (0.49, 1.96) | 0.97 | 0.96 (0.91, 1.02) | 0.183 | 1.01 (0.75, 1.35) | 0.97 |
| 574.12 | Cholelithiasis with other cholecystitis | 2650 | 17605642 | **1.12 (1, 1.25)** | **0.044** | 0.88 (0.69, 1.12) | 0.32 | 0.91 (0.48, 1.7) | 0.77 | 0.95 (0.76, 1.2) | 0.688 | 0.96 (0.61, 1.52) | 0.878 |
| 574.2 | Calculus of bile duct | 8401 | 17560937 | **1.07 (1, 1.15)** | **0.042** | 0.95 (0.78, 1.17) | 0.651 | 0.98 (0.46, 2.06) | 0.953 | 0.95 (0.87, 1.05) | 0.33 | 1.07 (0.94, 1.21) | 0.32 |
| 575.2 | Obstruction of bile duct | 1593 | 17626684 | **1.24 (1.1, 1.41)** | **<0.001** | 0.97 (0.28, 3.43) | 0.97 | 1.04 (0.18, 5.91) | 0.97 | **0.8 (0.7, 0.91)** | **0.001** | 1.1 (0.71, 1.69) | 0.688 |
| 578 | Gastrointestinal hemorrhage | 20111 | 17502200 | 0.96 (0.9, 1.03) | 0.296 | 0.94 (0.83, 1.08) | 0.393 | 0.91 (0.78, 1.07) | 0.249 | **1.08 (1.03, 1.12)** | **<0.001** | 1.01 (0.77, 1.33) | 0.93 |
| 578.8 | Hemorrhage of rectum and anus | 11095 | 17555331 | 0.96 (0.87, 1.06) | 0.413 | 0.93 (0.79, 1.1) | 0.395 | 0.92 (0.71, 1.18) | 0.505 | **1.09 (1.03, 1.15)** | **0.004** | 1 (0.8, 1.26) | 0.97 |
| **Genitourinary** | | | | | | | | | | | | | |
| 614.51 | Cervicitis and endocervicitis | 660 | 10488791 | 0.97 (0.52, 1.8) | 0.918 | 1.1 (0.7, 1.73) | 0.696 | **1.4 (1.03, 1.91)** | **0.033** | 0.93 (0.64, 1.34) | 0.706 | 1.16 (0.86, 1.57) | 0.326 |
| 622.2 | Mucous polyp of cervix | 1401 | 10498802 | **1.17 (1.03, 1.34)** | **0.02** | 0.99 (0.64, 1.53) | 0.97 | 1.12 (0.56, 2.21) | 0.766 | **0.83 (0.73, 0.95)** | **0.007** | 0.97 (0.43, 2.18) | 0.948 |
| 626.12 | Excessive or frequent menstruation | 10504 | 10375823 | **0.92 (0.86, 0.99)** | **0.026** | 0.96 (0.74, 1.24) | 0.757 | 1.02 (0.48, 2.14) | 0.97 | **1.1 (1.03, 1.17)** | **0.004** | 0.97 (0.76, 1.23) | 0.811 |
| **Pregnancy Complications** | | | | | | | | | | | | | |
| 634.3 | Ectopic pregnancy | 5034 | 10426967 | 0.97 (0.87, 1.08) | 0.592 | **1.11 (1.01, 1.21)** | **0.022** | 1.11 (0.93, 1.34) | 0.25 | 0.96 (0.86, 1.08) | 0.524 | 0.99 (0.65, 1.5) | 0.957 |
| 643 | Excessive vomiting in pregnancy | 4314 | 10470323 | **0.91 (0.85, 0.97)** | **0.005** | **1.17 (1.08, 1.27)** | **<0.001** | 1.12 (0.83, 1.51) | 0.48 | 1 (0.87, 1.14) | 0.97 | 1 (0.89, 1.11) | 0.97 |
| 643.1 | Hyperemesis gravidarum | 3558 | 10488446 | **0.9 (0.84, 0.96)** | **0.002** | 1.15 (0.97, 1.37) | 0.105 | 1.11 (0.91, 1.37) | 0.313 | 1.01 (0.76, 1.36) | 0.927 | 1.05 (0.78, 1.4) | 0.777 |
| 647.3 | Major puerperal infection | 274 | 10504689 | 1.04 (0.34, 3.13) | 0.952 | 1.37 (0.98, 1.92) | 0.068 | 0.78 (0.25, 2.45) | 0.688 | 0.86 (0.59, 1.25) | 0.439 | **0.72 (0.56, 0.93)** | **0.01** |
| 649.1 | Diabetes or abnormal glucose tolerance complicating pregnancy | 5053 | 10480682 | 0.92 (0.64, 1.33) | 0.679 | 1.22 (0.95, 1.56) | 0.118 | 1.17 (0.75, 1.82) | 0.504 | 0.95 (0.55, 1.64) | 0.866 | **1.26 (1.03, 1.55)** | **0.026** |
| 654.1 | Abnormality of organs and soft tissues of pelvis complicating pregnancy, childbirth, or the puerperium | 5842 | 10479533 | **0.94 (0.89, 0.99)** | **0.016** | **1.09 (1.01, 1.16)** | **0.019** | 0.93 (0.76, 1.13) | 0.467 | 1.04 (0.96, 1.12) | 0.378 | **1.08 (1.01, 1.16)** | **0.03** |
| 654.2 | Rhesus isoimmunization in pregnancy | 808 | 10503190 | 1.12 (0.91, 1.38) | 0.294 | 0.8 (0.61, 1.06) | 0.118 | 0.82 (0.4, 1.71) | 0.612 | 1.01 (0.65, 1.57) | 0.97 | **0.26 (0.24, 0.29)** | **<0.001** |
| 656.1 | Isoimmunization of fetus or newborn | 508 | 482914* | **2.6 (1.57, 4.33)\*\*** | **<0.001** | 1.57 (0.53, 4.61)** | 0.42 | 0.4 (0.01, 11.01)** | 0.599 | **0.24 (0.11, 0.51)\*\*** | **<0.001** | 1.03 (0.21, 5.13)** | 0.97 |
| **Dermatologic** | | | | | | | | | | | | | |
| 704 | Diseases of hair and hair follicles | 4102 | 17578353 | 0.99 (0.62, 1.59) | 0.97 | **1.18 (1.04, 1.34)** | **0.01** | 1.03 (0.25, 4.3) | 0.97 | 0.93 (0.82, 1.06) | 0.273 | 1.04 (0.68, 1.59) | 0.876 |
| 704.2 | Hirsutism | 1778 | 17608888 | 0.94 (0.7, 1.26) | 0.688 | **1.32 (1.15, 1.5)** | **<0.001** | 1.26 (0.96, 1.65) | 0.093 | 0.88 (0.75, 1.04) | 0.13 | 1.13 (0.87, 1.45) | 0.369 |
| 707 | Chronic ulcer of skin | 2121 | 17617533 | 0.94 (0.74, 1.19) | 0.617 | **1.23 (1.04, 1.47)** | **0.017** | 0.94 (0.29, 3.07) | 0.927 | 0.98 (0.51, 1.88) | 0.959 | 1.01 (0.59, 1.74) | 0.97 |

Table 2 continued on next page

Table 2 continued

| Phecode | Phenotype | Cases | Person-years | Blood group A IRR (95% CI) | p-value | Blood group B IRR (95% CI) | p-value | Blood group AB IRR (95% CI) | p-value | Blood group 0 IRR (95% CI) | p-value | Blood group RhD IRR (95% CI) | p-value |
|---|---|---|---|---|---|---|---|---|---|---|---|---|---|
| **Musculoskeletal** | | | | | | | | | | | | | |
| 722.7 | Intervertebral disc disorder with myelopathy | 747 | 17625461 | 0.95 (0.61, 1.47) | 0.82 | **1.26 (1.01, 1.59)** | **0.042** | 0.65 (0.31, 1.35) | 0.248 | 1.01 (0.58, 1.76) | 0.97 | 0.97 (0.28, 3.31) | 0.961 |
| 727 | Other disorders of synovium, tendon, and bursa | 30487 | 17272753 | 1.02 (0.95, 1.09) | 0.627 | **0.94 (0.89, 0.99)** | **0.016** | 0.99 (0.57, 1.7) | 0.97 | 1.01 (0.92, 1.11) | 0.809 | 0.97 (0.9, 1.04) | 0.352 |
| 727.7 | Contracture of tendon (sheath) | 391 | 17628006 | 0.91 (0.45, 1.84) | 0.8 | 1.14 (0.38, 3.43) | 0.826 | 1.11 (0.05, 24.01) | 0.951 | 1.02 (0.46, 2.25) | 0.97 | **0.65 (0.47, 0.9)** | **0.009** |
| 728 | Disorders of muscle, ligament, and fascia | 11338 | 17510763 | **1.08 (1.02, 1.13)** | **0.008** | 0.93 (0.83, 1.04) | 0.199 | 0.88 (0.74, 1.05) | 0.164 | 0.98 (0.85, 1.12) | 0.766 | 0.93 (0.85, 1.01) | 0.085 |
| 740 | Osteoarthrosis | 53711 | 17187748 | 1.01 (0.94, 1.09) | 0.815 | **0.95 (0.9, 0.99)** | **0.021** | 1.02 (0.79, 1.31) | 0.911 | 1.01 (0.93, 1.1) | 0.838 | 0.97 (0.92, 1.01) | 0.132 |
| 740.11 | Osteoarthrosis, localized, primary | 40386 | 17333061 | 1.01 (0.94, 1.09) | 0.779 | **0.94 (0.9, 0.99)** | **0.012** | 0.99 (0.72, 1.35) | 0.94 | 1.01 (0.95, 1.08) | 0.663 | 0.97 (0.92, 1.02) | 0.188 |
| 741 | Symptoms and disorders of the joints | 5085 | 17578450 | 0.99 (0.66, 1.48) | 0.97 | **0.85 (0.73, 0.99)** | **0.032** | 0.93 (0.52, 1.69) | 0.83 | 1.09 (0.99, 1.2) | 0.072 | 1 (0.82, 1.24) | 0.97 |
| 742 | Derangement of joint, non-traumatic | 16652 | 17438176 | 1.03 (0.96, 1.11) | 0.357 | **0.9 (0.83, 0.97)** | **0.004** | 0.98 (0.61, 1.56) | 0.924 | 1.02 (0.88, 1.17) | 0.838 | 0.95 (0.87, 1.04) | 0.291 |
| 742.9 | Other derangement of joint | 13669 | 17490182 | 1.02 (0.9, 1.16) | 0.757 | **0.91 (0.83, 0.99)** | **0.023** | 0.98 (0.47, 2.06) | 0.97 | 1.02 (0.91, 1.15) | 0.714 | 0.95 (0.86, 1.04) | 0.26 |
| 743 | Osteoporosis, osteopenia and pathological fracture | 15875 | 17536424 | **1.05 (1, 1.11)** | **0.049** | 0.94 (0.85, 1.05) | 0.26 | 0.92 (0.76, 1.12) | 0.405 | 0.99 (0.85, 1.14) | 0.858 | 1 (0.98, 1.02) | 0.97 |
| 743.11 | Osteoporosis NOS | 13633 | 17553976 | **1.06 (1.02, 1.1)** | **0.003** | **0.93 (0.87, 1)** | **0.039** | 0.94 (0.78, 1.13) | 0.514 | 0.98 (0.9, 1.07) | 0.68 | 1 (0.82, 1.21) | 0.97 |
| **Congenital Anomalies** | | | | | | | | | | | | | |
| 747 | Cardiac and circulatory congenital anomalies | 15297 | 482914* | **1.07 (1.01, 1.14)**** | **0.029** | 0.94 (0.8, 1.11)** | 0.475 | 1.02 (0.47, 2.23)** | 0.961 | 0.95 (0.88, 1.03)** | 0.223 | 0.98 (0.77, 1.25)** | 0.884 |
| 747.1 | Cardiac congenital anomalies | 1621 | 482914* | **1.19 (1.02, 1.39)**** | **0.023** | **0.77 (0.6, 0.98)**** | **0.036** | 1.03 (0.27, 3.86)** | 0.97 | 0.93 (0.69, 1.25)** | 0.635 | 1.02 (0.48, 2.16)** | 0.97 |
| 755 | Congenital anomalies of limbs | 6557 | 482914* | 0.99 (0.65, 1.5)** | 0.957 | 0.96 (0.71, 1.29)** | 0.778 | 0.96 (0.46, 1.97)** | 0.91 | 1.04 (0.89, 1.23)** | 0.627 | **0.88 (0.8, 0.97)**** | **0.013** |
| 755.61 | Congenital hip dysplasia and deformity | 2823 | 482914* | 1.01 (0.57, 1.78)** | 0.97 | 0.87 (0.66, 1.16)** | 0.355 | 0.89 (0.42, 1.88)** | 0.766 | 1.07 (0.85, 1.35)** | 0.57 | **0.81 (0.71, 0.94)**** | **0.004** |

a. Statistically significant IRRs are marked with bold (FDR adjusted *P*-value <0.05).

b. The IRRs are adjusted for age, sex, interaction between age and sex, and birth year.

c. All other ABO blood groups and the RhD negative blood group was used as a reference, respectively.

d. FDR-adjusted p-values and 95% confidence intervals are presented.

e. FDR adjusted p-values above 0.97 were set to 0.97 to avoid exploding adjusted confidence intervals.

e. Phecodes are divided by PheWAS disease categories.

f. The number of events and the follow-up time in person-years for each phecode is presented.

g. For study results of congenital phecodes estimates marked with ** are prevalence ratios instead of IRRs and the corresponding person-year marked with * are the size of the cohort.

The online version of this article includes the following source data for table 2:

**Source data 1.** Associations between the ABO/RhD blood groups and phecode incidence rate ratios.

example, respiratory tuberculosis (A16), tuberculosis of nervous system (A17), and miliary tuberculosis (A19), are combined into the phecode tuberculosis (phecode 10). Phecodes may therefore provide increased power and precision compared with using ICD-10 categories (*Denny et al., 2010*). Further, contrary to previous studies, we compared each blood group to all other blood groups, instead of determining effect estimates relative to blood group O. Thus, here we better capture the uniqueness of each individual ABO blood group.

## Limitations

Our study has some important limitations, firstly, the retrospective inclusion of patients and person-time may have introduced an immortal time bias from deaths before enrollment (in-hospital ABO/RhD blood group test) (*Yadav and Lewis, 2021*). The findings are therefore conditioned on patients surviving until the enrollment period. This implies, for example, that if a specific blood group causes

**Table 3.** Statistically significant associations between the ABO/RhD blood groups and the age of the first diagnosis.

| Phecode | Phenotype | N | Blood group A Estimate (95% CI) | p-value | Blood group B Estimate (95% CI) | p-value | Blood group AB Estimate (95% CI) | p-value | Blood group 0 Estimate (95% CI) | p-value | Blood group RhD Estimate (95% CI) | p-value |
|---|---|---|---|---|---|---|---|---|---|---|---|---|
| 079 | Viral infection | 25075 | −0.26 (−3.06, 2.54) | 0.864 | **0.92 (0.13, 1.71)** | **0.022** | 0.53 (−6.18, 7.23) | 0.887 | −0.23 (−3.22, 2.75) | 0.887 | 0.27 (−5.22, 5.75) | 0.93 |
| 451 | Phlebitis and thrombophlebitis | 16748 | **−0.58 (−1.02,−0.13)** | **0.011** | −0.24 (−5.71, 5.22) | 0.936 | −0.6 (−5.87, 4.66) | 0.833 | **0.91 (0.57, 1.25)** | **<0.001** | 0.01 (−0.33, 0.34) | 0.97 |
| 451.2 | Phlebitis and thrombophlebitis of lower extremities | 15650 | −0.53 (−1.08, 0.01) | 0.055 | −0.27 (−5.85, 5.31) | 0.93 | −0.7 (−6.35, 4.95) | 0.82 | **0.9 (0.55, 1.25)** | **<0.001** | 0.08 (−3.89, 4.06) | 0.97 |
| 474 | Acute and chronic tonsillitis | 41428 | −0.29 (−1.45, 0.87) | 0.634 | 0.42 (−1.97, 2.8) | 0.744 | 0.38 (−5.71, 6.48) | 0.909 | 0.05 (−2.15, 2.24) | 0.97 | **0.67 (0.15, 1.19)** | **0.011** |
| 474.1 | Acute tonsillitis | 18162 | 0.1 (−4.42, 4.61) | 0.97 | 0.41 (−7.17, 7.98) | 0.923 | 0.75 (−7.23, 8.74) | 0.864 | −0.42 (−3.76, 2.93) | 0.82 | **1.34 (0.64, 2.04)** | **<0.001** |

a. Statistically significant effect estimates are marked with bold (FDR adjusted *P*-value <0.05).

b. FDR adjusted p-values and 95% confidence intervals are presented.

c. FDR adjusted p-values above 0.97 were set to 0.97 to enable estimation of adjusted confidence intervals.

d. Estimates represent increases or decreases in years of age of first diagnosis.

The online version of this article includes the following source data for table 3:

**Source data 1.** Associations between the ABO/RhD blood groups and the age of the first diagnosis.

a higher incidence of a deadly disease, then patients with such blood group are more likely to have died before enrollment, and therefore fewer individuals having both that blood group and the disease will be present in our cohort. If so, the direction of the estimates for deadly diseases strongly related to any blood group will have been lowered or even flipped, relative to any causal relationship. The study design, however, enabled 41 year of follow-up and was deemed reasonable because the blood groups have not been associated with mortality differences. Moreover, the blood group distribution in our cohort was found to be almost identical to a reference population of 2.2 million Danish blood donors. Further, we replicated several findings of associations between the blood groups and severe diseases, including pancreatic cancer (*Vasan et al., 2016*; *Liumbruno and Franchini, 2014*). This may indicate that the potential bias was less prevalent. Further, by controlling for year of birth, the potential effects of immortal time bias were likely reduced, however, this could not be tested. Immortal time biases are potentially applicable in many biobanks studies, e.g. when using the UK Biobank for retrospective studies (*Yadav and Lewis, 2021*).

The generalizability of our findings is limited further because our cohort solely included hospitalized patients with known ABO and RhD blood groups. These are patients whom the treating doctor has deemed likely to potentially require a blood transfusion during hospitalization. The patients under study might therefore suffer from other diseases than patients without a determined blood group, and than never hospitalized individuals. Further, diseases that do not require hospitalization could not be examined. If the effect sizes are modified by factors which are more common in our cohort than in the general population then the estimates may not be generalizable. However, it is unclear if such effect modifier exists. Lastly, it was not possible to adjust for possible confounding from the geographical distribution or ethnicity of the patients (*Anstee, 2010*). This may have biased some estimates because the distribution of blood groups varies between ethnicities while ethnicity is also associated with differences in disease susceptibility. Particularly, ethnicity has been associated with differences in prevalence of infectious-, cardiovascular-, sickle cell disease, and thasalamia (*Kurian and Cardarelli, 2007*; *McQuillan et al., 2004*). Thus, the estimate of these disease groups should be interpreted with caution. The Danish population is however quite homogenous and approximately 94% of Danes have European ancestry (*Supplementary file 1*). Therefore, a potential bias from ethnicity may be less prevalent in our cohort as compared with studies in populations of more admixed origin.

In conclusion, we found the ABO/RhD blood groups to be associated with a wide spectrum of diseases, including cardiovascular-, infectious-, gastrointestinal- and musculoskeletal diseases. This may indicate that some of the potential selective pressure on the blood groups can be attributed to

disease susceptibility differences. We found few associations between the blood groups and age of first diagnosis.

## Acknowledgements

This study was performed as a part of the CAG (Clinical Academic Group) Center for Endotheliomics under the Greater Copenhagen Health Science Partners (GCHSP). Sources of Funding The study was supported by the Novo Nordisk Foundation (grants NNF14CC0001 and NNF17OC0027594) and the Innovation Fund Denmark (grant 5153-00002B). The funders played no role in the conduct of the study. Funding Novo Nordisk Foundation and the Innovation Fund Denmark

## Additional information

### Competing interests

Pär Ingemar Johansson: has received grants from the AP Møller Foundation, Innovation Fund Denmark and Novo Nordisk Foundation. The author has been issued the following patents: Publication no: 20110201553, 20110268732, 20130040898, 20130261177, 20150057325, 20160113891, 9381166, 9381243, 20160250164, 9433589, 20160303040 and US20090053193A1. PI Johansson reports ownership of stocks in Trial-Lab AB, Endothel Pharma ApS, TissueLink ApS, and MoxieLab ApS. PI Johansson declares that the financial interests listed have no impact on the submitted work. The author has no other competing interests to declare. The author declares that the financial interests listed have no impact on the submitted work. Søren Brunak: participates on the Danish National Genome Center advisory board and is the Chairman for the data infrastructure board. The author has stock in Intomics A/S, Hoba Therapeutics Aps, Novo Nordisk A/S, Lundbeck A/S and ALK Abello. The author participates on the board of directors for both Proscion A/S and Intomics A/S. The author has no other competing interests to declare. SB declares that the financial interests listed have no impact on the submitted work. The other authors declare that no competing interests exist.

### Funding

| Funder | Grant reference number | Author |
|---|---|---|
| Novo Nordisk Fonden | NNF14CC0001 | Søren Brunak |
| Novo Nordisk Fonden | NNF17OC0027594 | Søren Brunak |
| Innovation Fund Denmark | 5153-00002B | Søren Brunak |

The funders had no role in study design, data collection and interpretation, or the decision to submit the work for publication.

### Author contributions

Peter Bruun-Rasmussen, Conceptualization, Data curation, Software, Formal analysis, Validation, Investigation, Visualization, Methodology, Writing - original draft, Project administration, Writing – review and editing; Morten Hanefeld Dziegiel, Conceptualization, Writing – review and editing; Karina Banasik, Pär Ingemar Johansson, Conceptualization, Supervision, Writing – review and editing; Søren Brunak, Conceptualization, Resources, Supervision, Writing – review and editing

### Author ORCIDs

Peter Bruun-Rasmussen http://orcid.org/0000-0002-3595-1311
Morten Hanefeld Dziegiel http://orcid.org/0000-0001-8034-1523
Søren Brunak http://orcid.org/0000-0003-0316-5866

### Ethics

Human subjects: This is a register-based study and informed consent for such studies is waived by the Danish Data Protection Agency. Data access was approved by the Danish Patient Safety Authority (3-3013-1731), the Danish Data Protection Agency (DT SUND 2016-50 and 2017-57) and the Danish Health Data Authority (FSEID 00003092 and FSEID 00003724).

Decision letter and Author response
Decision letter https://doi.org/10.7554/eLife.83116.sa1
Author response https://doi.org/10.7554/eLife.83116.sa2

## Additional files

### Supplementary files

• Supplementary file 1. Population at the first day of the quarter by region and country of origin.

• Supplementary file 2. List of Phecodes defined as congenital or hereditary.

• Supplementary file 3. The study sample birth year distribution.

• Supplementary file 4. Associations between the ABO/RhD blood groups and Phecode incidence rate ratios of all analyzed Phecodes. a. Statistically significant IRRs are marked with bold (FDR adjusted *P*-value <0.05). b. The IRRs are adjusted for age, sex, interaction between age and sex, and birth year. c. All other ABO blood groups and the RhD negative blood group was used as a reference, respectively. d. The FDR adjusted p-values and 95% confidence intervals are presented. e. FDR adjusted p-values above 0.97 were set to 0.97 to avoid exploding adjusted confidence intervals. e. Phecodes are divided by PheWAS disease categories. f. The number of events and the follow-up time in person-years for each Phecode is also presented. g. For study results of congenital Phecodes estimates marked with ** are prevalence ratios instead of IRRs and the corresponding person-year marked with * are the size of the cohort.

• Supplementary file 5. Associations between ABO/RhD blood groups and the age of the first diagnosis. a. Statistically significant effect estimates are marked with bold (FDR adjusted *P*-value <0.05). b. The FDR adjusted p-values and 95% confidence intervals are presented. c. FDR adjusted p-values above 0.97 were set to 0.97 to enable estimation of adjusted confidence intervals. d. Estimates represent increases or decreases in years of age of first diagnosis.

• Supplementary file 6. Statistically significant associations for blood groups A, B and AB relative to blood group O. Further, also for RhD positive blood group relative to the RhD negative blood group. a. Statistically significant IRRs are marked with bold (FDR adjusted *P*-value <0.05). b. The IRRs are adjusted for age, sex, interaction between age and sex, and birth year. c. Blood group O and the RhD negative blood group was used as a reference, respectively. d. The FDR adjusted p-values and 95% confidence intervals are presented. e. FDR adjusted p-values above 0.97 were set to 0.97 to avoid exploding adjusted confidence intervals. f. Phecodes are divided by PheWAS disease categories. g. The number of events and the follow-up time in person-years for each Phecode is also presented. h. For study results of congenital Phecodes estimates marked with ** are prevalence ratios instead of IRRs and the corresponding person-year marked with * are the size of the cohort.

• Supplementary file 7. Associations between the ABO/RhD blood groups and Phecode incidence rate ratios of all analyzed Phecodes with blood group O and RhD negative as the reference, respectively. a. Statistically significant IRRs are marked with bold (FDR adjusted *P*-value <0.05). b. The IRRs are adjusted for age, sex, interaction between age and sex, and birth year. c. Blood group O and the RhD negative blood group was used as a reference, respectively. d. The FDR adjusted p-values and 95% confidence intervals are presented. e. FDR adjusted p-values above 0.97 were set to 0.97 to avoid exploding adjusted confidence intervals. f. Phecodes are divided by PheWAS disease categories. g. The number of events and the follow-up time in person-years for each Phecode is also presented. h. For study results of congenital Phecodes estimates marked with ** are prevalence ratios instead of IRRs and the corresponding person-year marked with * are the size of the cohort.

• MDAR checklist

### Data availability

Anonymized patient data was used in this study. Due to national and EU regulations, the data cannot be shared with the wider research community. However, data can be accessed upon relevant application to the Danish authorities. The Danish Patient Safety Authority and the Danish Health Data Authority have permitted the use of the data in this study; whilst currently, the appropriate authority for journal data use in research is the regional committee ("Regionsråd"). The statistical summary data used to create the tables and graphs are available as *Table 2—source data 1* and *Table 3—source data 1*. The analysis code is publicly available through https://www.github.com/peterbruun/blood_type_study (copy archived at *Bruun-Rasmussen, 2023*).

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
