## [Editor Report]

This important retrospective analysis of nearly 500,000 hospitalized Danish patients sheds light on the possible relationships between blood type and susceptibility to a host of diseases. The Danish National Patient Register is a compelling data source, and the statistical methodology is solid. The findings reported herein provide evidence, supporting information, and potential hypotheses for researchers interested in the causes and etiology of diseases as they relate to blood type.

---

## [Decision Letter]

**Decision letter after peer review:**

Thank you for submitting your article "Associations of ABO and Rhesus D Blood Groups with Phenome-Wide Disease Incidence: A 41-year Retrospective Cohort Study of 482,914 Patients" for consideration by *eLife*. Your article has been reviewed by 3 peer reviewers, including Philip Boonstra as Reviewing Editor and Reviewer #3, and the evaluation has been overseen by a Senior Editor.

Essential revisions:

1) Reviewers 1 and 3 both point out an issue regarding the A and O subgroups dominating the analyses due to their sample size. This seems to be a very important caveat to include in the comparison of the number of statistically significant findings per blood group.

2) Reviewer 1 comments on the lack of adjustment for patient ethnicity as a confounder (or a surrogate for other confounders). Please engage with this comment, which may involve explaining why this is unlikely, or which may involve actually trying to incorporate patient ethnicity into your models.

3) All of the reviewers raise many other good points in their Comments to the Authors, which I encourage you to read and engage with, potentially adjusting your analyses if you believe appropriate.

*Reviewer #1 (Recommendations for the authors):*

This study aims to address the important question of how different blood types are related to disease risk and age at diagnosis.

However, a major concern is that as per the comments in the public review, the lack of adjustment for confounding due to ethnicity represents a highly substantial limitation of this work. While it is briefly mentioned in the manuscript, this is a very major limitation that leads to very limited interpretability of the results. Incorporating ethnicity as a covariate into the analyses would be crucial.

*Reviewer #2 (Recommendations for the authors):*

Abstract/Intro

– "we determined the uniqueness" is a bit vague, could you more explicitly say you perform tests with A, AB, B, and O blood groups each as reference group as opposed to only O as the reference group?

– "diagnosis-wide" or "disease-wide" was used but perhaps more accurate to say "phenome wide" as in the title? Both disease-wide and diagnosis-wide are also used in the introduction before phecodes are introduced, and consistency might be better here.

– Age of disease onset specifically, not just disease onset, and perhaps age at first diagnosis is the most accurate (as is used in the introduction).

Methodological considerations

– ICD9 wasn't used? Most of the phecode mapping was done in ICD9/ICD10. Am I understanding ref 16 is in preparation? It would be important to describe how this is done and how it may bias your phecodes, particularly if ref 16 is not pre-printed yet. A good sanity check is how the prevalence/incidence of a handful of traits with this mapping compares to any other population-wide prevalence/incidence measures.

– It would be great to see a sensitivity analysis using either a mixed model to adjust for cryptic relatedness, close family structure, and population structure (presuming like other countries, the DNPR has many relatives). This may not statistically work with the quasi-Poisson model, so perhaps just restricting it to <3 degree individuals and comparing findings (of course this will decrease power, but nice to see if some of the main findings remain).

– I see it in the limitations, but please comment earlier in the manuscript on who gets a blood group determination in the hospital (e.g. people who made need a transplant sooner). Is it possible to characterize the disease prevalence in the subsection of the DNPR with a blood type and without so you can identify any major disease group biases?

– Was a power calculation used to determine the need for 100 cases?

– How is emigration recorded? National registers?

– The interquartile range would be a better descriptor of follow-up time rather than just the maximum (41 years) in the methods and in the limitations section.

– Why use a log-linear quasi-Poisson regression to estimate incidence rate ratios as opposed to logistic regression and odds ratios? It could be a valuable addition to the paper to provide odds ratios as well.

– It's good to adjust for ABO and RhD when testing RhD and ABO respectively, but is it possible to use interaction terms and consider these as well?

– It's great to use birth year to adjust for any "cohort effects" in society over time. Is attained age the age at end of the study period? Wouldn't birth year and attained age be too highly correlated to use both in the model? What is the rationale for using cubic splines for these variables rather than the numerical variables themselves? I see from the code that 20-year increments were used for the knots, any rationale for this methodology would be helpful.

– By excluding patients assigned a phecode at the start of the DNPR, would these be people who had the diagnosis previously and were recorded upon the inauguration of the register? Is there any kind of washout period you are using to define "start" of the DNPR?

– What is the ancestral breakdown of the cohort? Mostly European ancestries? While our current labels for genetic ancestry are quite rough, I think this is an important piece of information given the different distributions of blood groups across global populations.

– Excellent availability of code and summary data for tables/graphs.

Results

– The audience may be less interested in the number of significant phecodes and more in patterns. It could be good to comment on the shared phecodes between the 50, 38, 11, 53, and 28 found. What large-scale disease groupings (phewas disease categories) do these tend to fall in? Does one blood group have far more cardiovascular phecodes than another? Are any phecodes significant for more than 3 blood groups? Etc.

– Personally, I prefer p-values in scientific notation rather than <0.001 but I understand Table 2 is a lot of data to present.

– The figures could benefit from larger labels for readability.

– For the Manhattan plots, it would be good to specify -log10 FDR transformed adjusted p-values on the y-axis in addition to the figure legend.

– Were any blood groups associated with an earlier onset of outcomes?

Discussion

– What was used to identify "novel associations"? A systematic literature review? Comparison to Dahlén? I would refrain from using novel unless you define specifically how it was determined to be novel.

– A systematic comparison with what seems to be the closest study, Dahlén et al., would be beneficial as a type of replication.

– I would refrain from using the term linkage in the discussion as that may lead the reader to think of chromosomal linkage, but I think the authors mean a causal association.

– I don't think the findings support the discussion point on the selective pressure.

*Reviewer #3 (Recommendations for the authors):*

1. Could the authors justify why choosing to fit separate models comparing one blood type against all others, e.g. A vs. all others then AB vs. all others, is the more sensible choice than fitting one model that jointly tests for A vs. AB vs. B vs. O? I understand that there are various interpretative and statistical challenges to both, but fitting separate models is not internally consistent. The 'A vs. all others' model implicitly assumes that there is no difference in incidence in the AB, B, and O groups, but then the next model ('AB vs. all others') makes a different assumption, namely that there is no difference in incidence in the A, B, and O groups.

2. A natural limitation to this analysis is that there are more statistically significant findings in the O and A blood groups because they are the more prevalent groups, and statistical significance is driven by sample size. In this sense, it would be interesting if there were a way to account for the differences in sample size between the blood groups. Is it possible to investigate whether any of the groups have disproportionately more statistically significant findings after accounting for sample size?

3. Page 6, line 124: I think the use of the word 'confounder' here is not quite right in the technical sense, as I do not read this sentence to be claiming that sex is influencing blood type.

4. Regarding the legend for Figure 2:

a. It should have triangles rather than circles. Assuming this plot was made in ggplot2, this can be done using the override.aes argument in the guides function.

b. it would be helpful to show more than 3 values on the legend.

c. It would be helpful to use the same scale across the subfigures. What I mean is, in the bloodgroup AB figure, there is no discernible difference in size between a 1.1 and a 4.0 rate ratio.

d. I realize this is very pedantic but I believe the legend is technically not showing rate ratios but rather max(rate_ratio, 1/rate_ratio).

5. Do the authors have any intuition why Figure 1 is bimodal? My interpretation of this figure is that, among those who were hospitalized in Denmark between 2006 and 2018, the plurality was born either in the immediate post-WW2 era (makes sense to me) or the 80s (doesn't make as much sense to me).

6. Page 7, line 152: reference 20 is not related to FDR. Can the authors provide a reference for their specific approach to controlling FDR?

---

## [Author Response]

Essential revisions:1) Reviewers 1 and 3 both point out an issue regarding the A and O subgroups dominating the analyses due to their sample size. This seems to be a very important caveat to include in the comparison of the number of statistically significant findings per blood group.

This has now been mentioned in the lines where the number of statistically significant findings per blood groups are mentioned (lines 191-194). Further, a supplemental analysis has been included where blood group O is used as the reference (Supplementary files 6–7.

2) Reviewer 1 comments on the lack of adjustment for patient ethnicity as a confounder (or a surrogate for other confounders). Please engage with this comment, which may involve explaining why this is unlikely, or which may involve actually trying to incorporate patient ethnicity into your models.

We unfortunately do not have access to information on ethnicity. We have further elaborated on this limitation in the manuscript and pointed out that the Danish population is quite homogenous why this is less of a concern in our study as compared with studies of populations with more admixed origin (lines 341-349). For a more detailed discussion on this please see our response to the reviewers’ comments.

3) All of the reviewers raise many other good points in their Comments to the Authors, which I encourage you to read and engage with, potentially adjusting your analyses if you believe appropriate.

We have engaged with all the comments and have changed the manuscript and figures accordingly.

Reviewer #1 (Recommendations for the authors):This study aims to address the important question of how different blood types are related to disease risk and age at diagnosis.However, a major concern is that as per the comments in the public review, the lack of adjustment for confounding due to ethnicity represents a highly substantial limitation of this work. While it is briefly mentioned in the manuscript, this is a very major limitation that leads to very limited interpretability of the results. Incorporating ethnicity as a covariate into the analyses would be crucial.

We agreed that ethnicity may introduce bias in the analysis of some diseases. We have now further elaborated on this limitation in lines 341-349. Unfortunately, we do not have information on ethnicity, and therefore an analysis adjusting for ethnicity is not possible with the available data. While we agree that ethnicity is a concern, it should be noted that ethnicity will only introduce confounding for the analysis of diagnosis which are more common for certain ethnicities and where at the same time the distribution of the ABO/RhD blood groups are different between ethnicities. This may e.g. be the case for infectious diseases, cardiovascular diseases and bleeding disorders as has now been mentioned in the limitations. Further, the Danish population is very homogeneous (approximately 94% have European origin). Therefore, a potential bias from not being able to adjust for ethnicity will be less distinct in our study. In Supplement Figure 1, we have now provided a graph showing the ancestry of the complete Danish population, the Capital Region of Denmark, and Region Zealand from 2008-2016 (Supplemenary file 1). From the graph it can be seen that approximately 90% of the population in the Capital Region and 97% of the Region Zealand population were of European origin. This information has now been added in the method section lines 99-102.

Reviewer #2 (Recommendations for the authors):Abstract/Intro– "we determined the uniqueness" is a bit vague, could you more explicitly say you perform tests with A, AB, B, and O blood groups each as reference group as opposed to only O as the reference group?

The formulation has now been changed to “we determined the incidence rate ratios for each individual ABO blood group relative to all other ABO blood groups…” (lines 37-38).

– "diagnosis-wide" or "disease-wide" was used but perhaps more accurate to say "phenome wide" as in the title? Both disease-wide and diagnosis-wide are also used in the introduction before phecodes are introduced, and consistency might be better here.

We have chosen not to use the term “phenome wide” in the manuscript text because “phenome wide” without the mentioning of disease is to be understood as a PheWAS analysis where genetic information of a specific SNP is used. The title is combining “phenome-wide” with “disease incidence” to underline that it is not a classic PheWAS study. As suggested by the reviewer we have now consistently used “disease-wide” in the introduction.

– Age of disease onset specifically, not just disease onset, and perhaps age at first diagnosis is the most accurate (as is used in the introduction).

We agree that age at first diagnosis is the most accurate formulation. This has now been changed throughout the manuscript.

Methodological considerations– ICD9 wasn't used? Most of the phecode mapping was done in ICD9/ICD10. Am I understanding ref 16 is in preparation? It would be important to describe how this is done and how it may bias your phecodes, particularly if ref 16 is not pre-printed yet. A good sanity check is how the prevalence/incidence of a handful of traits with this mapping compares to any other population-wide prevalence/incidence measures.

ICD9 codes was never used in Denmark. Ref 16 is currently under review. I have received a copy of the manuscript by the authors which I have now uploaded as a “related manuscript file” for the reviewers to see. The authors do unfortunately not wish to make it available as preprint.

– It would be great to see a sensitivity analysis using either a mixed model to adjust for cryptic relatedness, close family structure, and population structure (presuming like other countries, the DNPR has many relatives). This may not statistically work with the quasi-Poisson model, so perhaps just restricting it to <3 degree individuals and comparing findings (of course this will decrease power, but nice to see if some of the main findings remain).

This is indeed an interesting suggestion. Unfortunately, we do not have access to information on population or family structure. However, we do not believe that family structure would impose a significant amount of bias given the large sample size. As mentioned in the limitation’s ethnicity may however impose a bias which have now been further elaborated in the limitations section.

– I see it in the limitations, but please comment earlier in the manuscript on who gets a blood group determination in the hospital (e.g. people who made need a transplant sooner). Is it possible to characterize the disease prevalence in the subsection of the DNPR with a blood type and without so you can identify any major disease group biases?

We have now commented earlier on in the manuscript on the kinds of patients which were included in the study (lines 99-100). Further, we have in the limitation elaborated on what this means for generalizability (lines 336-339). With the available data is it unfortunately not possible to identify any major disease group differences between the hospitalized patients with a blood group determination and those without.

– Was a power calculation used to determine the need for 100 cases?

A power calculation was not used. Similarly to the study by Dahlén et al. the number of cut-off cases was arbitrarily picked. We have now changed the formulation in line 111-112 so that is cannot be understood as if a power calculation had been done for each of the 5x1300 analysis.

– How is emigration recorded? National registers?

Yes. This information has now been added to the manuscript (line 104).

– The interquartile range would be a better descriptor of follow-up time rather than just the maximum (41 years) in the methods and in the limitations section.

In the methods and limitation we referred to the length of the study period which is the maximum length of follow-up. The median and interquartile range of follow-up time in the cohort is given in Table 1.

– Why use a log-linear quasi-Poisson regression to estimate incidence rate ratios as opposed to logistic regression and odds ratios? It could be a valuable addition to the paper to provide odds ratios as well.

A logistic regression would not take time-to-event into account. Thus, incorporating person-time using a log-linear Poisson regression enriches the analysis as compared with a logistic regression.

– It's good to adjust for ABO and RhD when testing RhD and ABO respectively, but is it possible to use interaction terms and consider these as well?

Similar to the study by Dahlén et al. we chose not to consider interactions between the ABO and RhD blood groups as there is little evidence that such interaction exists.

– It's great to use birth year to adjust for any "cohort effects" in society over time. Is attained age the age at end of the study period? Wouldn't birth year and attained age be too highly correlated to use both in the model? What is the rationale for using cubic splines for these variables rather than the numerical variables themselves? I see from the code that 20-year increments were used for the knots, any rationale for this methodology would be helpful.

Attained age is not the age at the end of the study period. If so attained age and birth year would indeed be highly correlated. Attained age is the underlying person-time. In a classic logistic regression time is not incorporated. However, we are doing a time-to-event study using a log-linear Possion regression incorporating person-time. Attained age is thus the age of each individual at each time period in the study. Thus, we are comparing individuals of the same age over time. As explained in the methods section, attained age is divide into 1-year time intervals and used as the underlying time.

We did use the numerical variables in the models however instead of assuming linear relationships we used cubic splines to allow flexible modeling which better adjusts for potential non-linear relationships. For a more elaborated discussion on the benefits modelling using restricted cubic splines the reviewer is referred to: Gauthier, J., Q. V. Wu, and T. A. Gooley. "Cubic splines to model relationships between continuous variables and outcomes: a guide for clinicians." *Bone marrow transplantation* 55.4 (2020): 675-680.

– By excluding patients assigned a phecode at the start of the DNPR, would these be people who had the diagnosis previously and were recorded upon the inauguration of the register? Is there any kind of washout period you are using to define "start" of the DNPR?

This is likely to be people who had the diagnosis previously which was then registered at time the registry was created. Few is also likely to be individuals who got the diagnosis at the very start of the registry. Because this is unknown, we chose to exclude these patients from the analysis as this information was regarded too noisy. There is no wash-out period. The “start” of the DNPR is defined as the official start of the registry. Few patient were excluded from this procedure and thus the effect on the analysis is very limited.

– What is the ancestral breakdown of the cohort? Mostly European ancestries? While our current labels for genetic ancestry are quite rough, I think this is an important piece of information given the different distributions of blood groups across global populations.

The Danish population is very homogeneous and mostly of European ancestry. We have now added a table in Supplementary file 1 showing the ancestral distribution in Denmark in the inclusion period. We do unfortunately not have any information available on ancestry or ethnicity of the study sample. This fact has now been further elaborated in the limitations section lines 341-348.

– Excellent availability of code and summary data for tables/graphs.

Thanks. We have tried to make the analysis code easy accessible to allow for replication in other cohorts.

Results– The audience may be less interested in the number of significant phecodes and more in patterns. It could be good to comment on the shared phecodes between the 50, 38, 11, 53, and 28 found. What large-scale disease groupings (phewas disease categories) do these tend to fall in? Does one blood group have far more cardiovascular phecodes than another? Are any phecodes significant for more than 3 blood groups? Etc.

We have now commented on the phewas disease groupings of the blood groups (lines 198-201).

– Personally, I prefer p-values in scientific notation rather than <0.001 but I understand Table 2 is a lot of data to present.

This was also something we considered. However, we decided that we would rather direct the reader’s attention to the confidence intervals than the p-value. Therefore, we used the <0.001 notation.

– The figures could benefit from larger labels for readability.

Larger labels have now been added to the figures.

– For the Manhattan plots, it would be good to specify -log10 FDR transformed adjusted p-values on the y-axis in addition to the figure legend.

This has now been specified on the y-axis.

– Were any blood groups associated with an earlier onset of outcomes?

In table 3 we show the findings from the analysis on the ABO and RhD blood groups association with age at the first diagnosis.

Discussion– What was used to identify "novel associations"? A systematic literature review? Comparison to Dahlén? I would refrain from using novel unless you define specifically how it was determined to be novel.

We did a literature search ourselves but agree that without a reference to a published systematic literature review the term novel should not be used. We have now removed the term “novel” lines 266 and 268.

– A systematic comparison with what seems to be the closest study, Dahlén et al., would be beneficial as a type of replication.

We agree with the reviewer that such comparison would be beneficial. However, a systematic comparison would be very comprehensive, and we believe it would be more appropriate as a manuscript on its own. In the discussion we have instead chosen to discuss a few selected findings some of which have also been found in the study by Dahlén et al. as indicated by a reference to the study by Dahlén et al. For example, we highlight associations of pancreatic cancer, gastroduodenal ulcers, type 2 diabetes, and cardiovascular diseases which was also observed by Dahlén et al. Lastly, it should be noted that the two studies cannot be compared directly as Dahlén et al. used blood groups O as the reference and we used all other blood groups as the reference.

– I would refrain from using the term linkage in the discussion as that may lead the reader to think of chromosomal linkage, but I think the authors mean a causal association.

The term linkage has now been changed to relationship (line 273).

– I don't think the findings support the discussion point on the selective pressure.

We agree, this point has now been removed from the discussion and conclusion.

Reviewer #3 (Recommendations for the authors):1. Could the authors justify why choosing to fit separate models comparing one blood type against all others, e.g. A vs. all others then AB vs. all others, is the more sensible choice than fitting one model that jointly tests for A vs. AB vs. B vs. O? I understand that there are various interpretative and statistical challenges to both, but fitting separate models is not internally consistent. The 'A vs. all others' model implicitly assumes that there is no difference in incidence in the AB, B, and O groups, but then the next model ('AB vs. all others') makes a different assumption, namely that there is no difference in incidence in the A, B, and O groups.

The aim of the study was to determine how each individual ABO blood group is distinct from all other ABO blood groups in terms of disease susceptibility. This question can only be answered by fitting separate models defining the patients as either having the blood group under study or not. An alternative, which would answer a different research question, would be to do a pairwise comparison between each ABO blood group would require 8x1,312 analyses (see response 1 to reviewer #1s comment). We do not believe that one model which jointly tests for all pair-wise comparisons can be fitted, the model would need to define a reference which in previous studies have been blood group O. The research question which has been asked in previous studies is thus how does blood group A, B and AB differ from blood group O in terms of disease susceptibility, respectively. We believe it to be more informative to ask how each blood group differs from all the other blood groups. However, we agree with the reviewer that this has its limitations because of the difference in the frequency of the individual ABO blood groups. We have now added a Supplementary Analysis where blood group O has been used as the reference (lines 145-147). The results of the supplemental analysis are presented in Supplementary files 6–7.

2. A natural limitation to this analysis is that there are more statistically significant findings in the O and A blood groups because they are the more prevalent groups, and statistical significance is driven by sample size. In this sense, it would be interesting if there were a way to account for the differences in sample size between the blood groups. Is it possible to investigate whether any of the groups have disproportionately more statistically significant findings after accounting for sample size?

We agree that this is a natural limitation to any study concerning blood groups and we have now commented on this in line 192-195. The only way to be able to better find associations for the less prevalent blood groups would be to include more patients, potentially by conducting a similar study combining cohorts from several countries to increase the power. We do not know of a way to determine if any of the blood groups have disproportionately more statistically significant findings while accounting for sample size.

3. Page 6, line 124: I think the use of the word 'confounder' here is not quite right in the technical sense, as I do not read this sentence to be claiming that sex is influencing blood type.

We agree with the reviewer that this was an incorrect formulation. We have now changed the statement.

4. Regarding the legend for Figure 2:a. It should have triangles rather than circles. Assuming this plot was made in ggplot2, this can be done using the override.aes argument in the guides function.

We thank the reviewer for this tip. The circles have now been replaced by triangles.

b. it would be helpful to show more than 3 values on the legend.

We have now added two more legends so that five values are shown. However, for the comparison of the IRRs the readers should instead use table 2.

c. It would be helpful to use the same scale across the subfigures. What I mean is, in the bloodgroup AB figure, there is no discernible difference in size between a 1.1 and a 4.0 rate ratio.

We have now applied the same legends in all plots.

d. I realize this is very pedantic but I believe the legend is technically not showing rate ratios but rather max(rate_ratio, 1/rate_ratio).

This is correct. It is showing the rate ratio if rate ratio >= 1 and 1/rate ratio if the rate ratio <1. This makes it possible to compare the sizes of the rate ratios for both positive and negative associations. The direction of the triangles shows if it is a positive or inverse association as mentioned in the figure text.

5. Do the authors have any intuition why Figure 1 is bimodal? My interpretation of this figure is that, among those who were hospitalized in Denmark between 2006 and 2018, the plurality was born either in the immediate post-WW2 era (makes sense to me) or the 80s (doesn't make as much sense to me).

We agree with the reviewers first point. The reason that the distribution is bimodal is likely because the older population will be hospitalized in the inclusion period corresponding to the first peak. The second peak is likely caused by pregnant women who would be hospitalized when given birth and who commonly have their blood type determined in the hospital.

6. Page 7, line 152: reference 20 is not related to FDR. Can the authors provide a reference for their specific approach to controlling FDR?

Thanks for spotting this error. We have now also added a reference to the specific approach used to control for FDR.